# When Storms Stir the Mediterranean Depths: Chlorophyll-a Response to Mediterranean Cyclones

Giovanni Scardino<sup>1,2\*</sup>, Alok Kushabaha<sup>1,3</sup>, Mario Marcello Miglietta<sup>4</sup>, Davide Bonaldo<sup>5</sup>, Giovanni Scicchitano<sup>1,2</sup>

- 1-Department of Earth and Geoenvironmental Sciences, University of Bari Aldo Moro, 70125 Bari, Italy
- 2- Interdepartmental Research Center for Coastal Dynamics, University of Bari Aldo Moro, 70125 Bari, Italy
- 3 IUSS School for Advanced Studies, Pavia, Italy
- 0 4 Institute of Atmospheric Sciences and Climate (CNR-ISAC), National Research Council of Italy, Padua, Italy
  - 5 Institute of Marine Sciences (CNR-ISMAR), National Research Council of Italy, Venice, Italy

Correspondence to: Giovanni Scardino (giovanni scardino @uniba.it)

Abstract. Mediterranean cyclones induce significant biogeochemical perturbations in the Mediterranean Sea, with particularly notable effects on chlorophyll-a (Chl-a) dynamics. This study examines Chl-a variability during twenty Mediterranean cyclones, comparing offshore and nearshore responses. Through an integrated investigation of reanalysis products, ARGO float observations, and Sentinel-2 multispectral satellite imagery, we quantified vertical and surface Chl-a concentrations, while concurrently assessing nitrate distribution, currents, and mixed layer depth (MLD) variability. Our results revealed that both Tropical-like Cyclones and Extratropical Cyclones generated a pronounced uplift of the deep chlorophyll maximum (DCM) in cyclones exhibiting slow-moving phases. Notably, wind-driven upwelling and air-sea heat exchange critically govern DCM uplift for these cyclones. We demonstrated that these physical mechanisms collectively drive DCM uplift along the trajectories of intense, slow-moving Mediterranean cyclones, with significant implications for nutrient cycling and primary productivity across the Mediterranean basin.

Keywords: chlorophyll; deep chlorophyll maximum; cyclones; shoaling; deepening

#### 1. Introduction

The dynamics of biogeochemical parameters in the oceans are influenced by extreme weather events. In particular, changes in chlorophyll and phytoplankton concentrations have been widely studied to better understand upper-ocean physical properties in response to severe weather (e.g. Gallisai et al., 2016; Karagiorgos et al., 2023). Among the intense weather systems in the Mediterranean basin, Tropical-like Cyclones (TLCs) and extratropical cyclones (ECs) are key drivers of alterations in the upper-ocean physical properties (Kotta, 2023; Kotta and Kitsiou, 2019b, a). Mediterranean TLCs (also known as medicanes; Miglietta et al., 2025)) represent weather systems characterized by the presence of a warm core due to air-sea interaction processes and the release of latent heat, similarly to tropical cyclones, thus differing from cold core ECs (Miglietta, 2019; Miglietta and Rotunno, 2019; Panegrossi et al., 2023). Both medicanes and intense ECs determine physical and biological oceanographic perturbations through intense winds and heavy rainfall (Jangir et al., 2023, 2024). Their

As for tropical cyclones, a decrease in SST is observed after TLC passage (Avolio et al., 2024; Scardino et al., 2024), due to upwelling, vertical mixing, together with cooling due to heat fluxes towards the atmosphere (Kassis and Varlas, 2021; Menna et al., 2023; Price, 1981; Sanford et al., 1987). Vertical mixing (in particular at the base of the mixed layer), upwelling, and divergent geostrophic currents both cause shoaling of the mixed layer depth (MLD) in the water column corresponding to the location of cyclone center (Piontkovski and Al-Hashmi, 2014). This, in turn, induces the upward transport of nutrients, favouring high biological productivity (Kotta and Kitsiou, 2019a, b; Piontkovski and Al-Hashmi, 2014). When these nutrients reach the sunlit euphotic zone —combined with adequate light for photosynthesis—chlorophylla (Chl-a) concentrations increase significantly (Latha et al., 2015). Generally, strong, slow-moving cyclones can induce significant sea surface cooling and phytoplankton blooms (Lin, 2012; Mei et al., 2015; Wang, 2020; Zhao et al., 2008).

These intense weather systems can also produce prolonged precipitation lasting several days (Owen et al., 2021; Tartaglione et al., 2006), altering the Chl-a concentrations in the Mediterranean Sea (Katara et al., 2008; Kotta et al., 2017).

The Mediterranean Sea is defined as an oligotrophic basin, with generally low nutrient availability and consequently low phytoplankton biomass (D'Ortenzio and Ribera d'Alcalà, 2009). Chl-a, a proxy for phytoplankton biomass, exhibits strong spatial and temporal variability due to the interplay of physical, chemical, and biological processes (Mélin et al., 2017). In the Mediterranean open sea, low Chl-a concentrations (0.05–0.3 mg/m³) are typically observed in the first meters of the water column, due to nutrient limitation (Li et al., 2024b; Marty and Chiavérini, 2010, 2002; Teruzzi et al., 2021). However, the coastal zones and river-influenced regions (e.g., near the Nile Delta, Rhône, or Po River outflows) exhibit higher Chl-a concentrations (0.5–5 mg/m³) at the surface (**Fig. 1**), due to terrestrial nutrient inputs (Ludwig et al., 2009).



While warming of the Mediterranean Sea surface appears to be a consistent pattern (Belkin, 2009; Rixen et al., 2005; Skliris et al., 2012), some uncertainties regard the phytoplankton dynamics, in particular in relation to meteorological forcings. Chla changes due to the occurrence of Mediterranean cyclones are not well understood yet. Other severe weather events may affect the dynamics of nutrients, Chl-a and deep chlorophyll maximum (DCM) differently, depending on the specific weather system (Kotta, 2023; Kotta and Kitsiou, 2019b; Walker et al., 2005).

A defining characteristic of phytoplankton dynamics in the Mediterranean Sea is the presence of a DCM, which typically forms near the base of the euphotic zone, aligned with the upper nutricline in both permanently and seasonally stratified waters (Cullen, 2015). Below the nutricline depth (more than 120 m), the concentration of nitrate and phosphate increases (Lazzari et al., 2016; Mignot et al., 2019). Satellite-derived data revealed an increase in the Chl-a concentration after the passage of medicanes, as observed after medicane Zeo (12-15 December 2005 (Kotta and Kitsiou, 2019b)) and medicane Ianos (15-20 September 2020 (Kotta, 2023)). Menna et al. (2023) observed an uplift of DCM during the occurrence of medicane Apollo (25 October-1 November 2021), concomitant with a shoaling of the nutricline and MLD. They described a sort of dome shape near the cyclone eye, where all vertical profiles of biogeochemical parameters were shoaled. Generally, the shoaling of biogeochemical parameters during the occurrence of strong Mediterranean cyclones seems to be associated with subsurface water mixing and wind-driven upwelling (Jangir et al., 2024; Mourre et al., 2023; Zhang et al., 2021).

Jangir et al. (2024) highlighted increased Chl-a and phytoplankton concentrations at the warm-core eddy locations for medicane Apollo. However, Kassis and Varlas (2021) reported a deepening in the MLD during the occurrence of medicane Zorbas (26-29 September 2018), caused by the inflow of Atlantic currents. Nonetheless, this event experienced an increase in Chl-a and phytoplankton concentration by approximately 0.01 mg/m<sup>3</sup> and 0.02 mmol/m<sup>3</sup>, respectively (Jangir et al., 2024).

Figure 1. Chlorophyll concentration in the Mediterranean basin extracted from Copernicus yearly dataset (in the map data refer to 2022); a) surface Chl-a concentration; b) Chl-a concentration extracted below the nutricline water depth (i.e. Chl-a concentration at 120 m). The areas where sea depth is below 120 m are in cyan. Figure contains the background maps provided by Esri World Ocean Basemap (sourced from *Esri's ArcGIS Online services*.).



In this study, we analysed the Chl-a variations throughout the water column and at the sea surface during intense TLCs and ECs. We selected twenty cyclones based on their intensity, estimated using mean sea level pressure (MSLP) and heat flux values from the ERA5 and Copernicus reanalysis dataset. Quasi-stationarity and slow-movement phases were assessed by tracking the MSLP minima for each event. Using remote sensing data and reanalysis products, we evaluated Chl-a concentrations in areas affected by slow-moving cyclone phases and along the impacted coastal regions. Subsequently, we correlated these variations with key biogeochemical parameters, including nitrate distribution, currents, MLD, and downward heat flux. Although previous studies have analysed a similar problem in other basins, this is the first time the issue has been analyzed in a systematic way in a peculiar basin as the Mediterranean.

The paper is organized as follows. Section 2 describes the materials and methods used to analyse Chl-a changes during Mediterranean cyclones. In particular, we assess the relationship between the translation speed and intensity of cyclones and their impact on Chl-a variations. Section 3 presents the results of our analysis, highlighting the uplift of the DCM and its association with changes in biogeochemical parameters, such as nitrate concentrations, current distributions, and MLD deepening/shoaling. Section 4 discusses the correlation between DCM uplift and physical parameters during different cyclone events, with a particular focus on DCM dynamics in the cyclone's center and periphery. The main findings are reported in Section 5.

## 2. Material and Methods: Analysis of Chlorophyll-a Changes During Mediterranean Cyclones

In this study, we analysed multiple datasets to reconstruct Chl-a dynamics in response to Mediterranean cyclone forcing. The following steps were undertaken:

- Assessment of slow-moving phases and heat energy seaward of the cyclones;
- Analysis of Chl-a dynamics in offshore and nearshore regions;
- Joint analysis of Chl-a dynamics with biogeochemical parameters.

To capture the full temporal evolution of biogeochemical parameters during the cyclone passage, we analyzed Sentinel-2 and Argo data alongside ERA5 (hourly data) and CMEMS (daily data) reanalysis products. This approach provided a comprehensive overview of the event's progression, particularly revealing time-series variations in mixed layer depth (MLD) and deep chlorophyll maximum (DCM) dynamics.

# 2.1 Identifying the areas influenced by Mediterranean cyclones and associated Chl-a changes

To identify the cyclone-impacted regions and the movement of the cyclones, we extracted hourly minimum mean sea-level pressure (MSLP) values from ERA5 reanalysis data (0.25 x 0.25 degree resolution) for the cyclones listed in **Table 1**. The cyclone-impacted areas were selected following the literature works, focusing on those areas greatly affected by wind stress (**Table 1**). The selected areas for each cyclone are reported in the Supplementary Material S1. Slow-moving phases were evaluated by tracking the displacement of hourly MSLP minima. According to Piontkovski and Al-Hashmi (2014), slow cyclones, characterized by a translation speed lower than 10 m/s, are able to induce Chl-a changes. Events exhibiting displacements shorter than 36 km over 4 hours were considered as slow-moving cyclone phases (these phases are underlined by the red shaded areas reported in **Fig. 2** for the case of medicane Zorbas). While, quasi-stationary phases have been considered for the displacements shorter than 10 km over 4 hours.

Then, we analysed the MSLP minima alongside daily sea surface heat fluxes (*hfds*), extracted from Copernicus reanalysis (Clementi et al., 2019; Escudier et al., 2020) at 0.04x0.04 degrees resolution to map the regions where air-sea interaction is more intense and may influence the cyclone dynamics.

Wind data from ERA5 were extracted in the following two components: the eastward wind component (U wind) and the northward wind component (V wind). These components were combined to obtain the wind speed during storm events as follows (Eq. 1):

*Wind speed* =  $\sqrt{U^2 + V^2}$  (Eq. 1)

To determine the cyclone-impacted area, we first identified the location of the maximum 10-meter wind speed within a 200 km radius of the cyclone center. The radius of maximum wind was then defined as the distance from the center to this location. The cyclone-impacted area was subsequently assessed as a circular area with the radius of maximum wind (see **Supplementary Material S1**).

Figure 2. Assessment of the stationarity phase of a cyclone from MSLP data (case of medicane Zorbas); a) extraction of minimum MSLP point with value of center (minimum) pressure and the maximum 10-meter wind speed within a 200 km radius from the cyclone center (gray shading areas); b) center pressure along the cyclone track and displacements assessed across each minimum MSLP point, red bars indicate the displacement when it was shorter than 36 km in 4 hours, while shaded red areas indicate the long lasting slow-moving phases.

The cyclones were analysed to assess the combined influence of their persistence and their associated intensity in a given area. To determine the total energy changes during cyclone events, a double integral was applied to the *hfds* within cyclone-affected areas:

$$Q_{total}(t) = \iint_A hfds(x, y, t)dx dy$$
 (Eq. 2)



$$E_{total} = \int_{t1}^{t2} Q_{total}(t) dt \quad \text{(Eq. 3)}$$

 $E_{total} = \int_{t1}^{t2} Q_{total}(t) dt$  (Eq. 3)  $Q_{total}(t)$  is equal to the total heat flux over area A at time t (units: W). The Eq. 2 provides the instantaneous power (energy per second) entering or leaving the Mediterranean Sea over a given region (specifically the areas affected by Chl-a changes).

$E_{total}$  represents the total energy change over time (units: J). Eq. 3 provides the net heat gained or lost by the ocean region during cyclone lifetime, i.e., between t1 (here 1 day before cyclogenesis) and t2 (here 1 day after cyclolysis).

Table 1. Mediterranean cyclones considered in this study for the analysis of Chl-a dynamics; classification is based

| Cyclone       | Dates of      | Classification | Features                         | Reference               | Impacted areas       |
|---------------|---------------|----------------|----------------------------------|-------------------------|----------------------|
|               | occurrence    |                |                                  |                         |                      |
| Zeo – 2005    | 12-15         | Mediterranean  | Deep warm core; flash floods     | (Fita et al., 2007;     | Eastern              |
|               | December 2005 | hurricane      |                                  | Gutiérrez-Fernández     | Mediterranean        |
|               |               |                |                                  | et al., 2024; Kotta     | (Cyprus, Lebanon)    |
|               |               |                |                                  | and Kitsiou, 2019b;     |                      |
|               |               |                |                                  | Miglietta and           |                      |
|               |               |                |                                  | Rotunno, 2019)          |                      |
| Janet - 2007  | 13-19 October | Extratropical  |                                  | (Flaounas et al.,       | Libya                |
|               | 2007          | cyclone        |                                  | 2022)                   |                      |
| Rolf – 2011   | 6-9 November  | Mediterranean  | Eye-like structure;              | (Cavicchia et al.,      | Gulf of Lyon         |
|               | 2011          | hurricane      | 6 November 2011: asymmetric deep | 2014; Ricchi et al.,    | (France), Corsica,   |
|               |               |                | cold core;                       | 2017; de la Vara et     | northwestern Italy   |
|               |               |                | 7-8 November 2011: deep warm     | al., 2021)              |                      |
|               |               |                | core                             |                         |                      |
| Akle - 2011   | 2-3 January   | Mediterranean  | Shallow warm core                | (Flaounas et al.,       | Southeastern Sicily, |
|               | 2011          | Tropical Storm |                                  | 2022, 2023)             | Greece               |
| Qendresa -    | 7-9 November  | Mediterranean  | Rapid intensification; >100 mm   | (Carrió, 2023;          | Malta, Sicily,       |
| 2014          | 2014          | hurricane      | rainfall in Malta                | Pytharoulis, 2018)      | eastern Tunisia      |
| Xandra –      | 1-4 December  | Mediterranean  | Weak warm core                   | (Di Muzio et al.,       | Western              |
| 2014          | 2014          | Tropical Storm |                                  | 2019)                   | Mediterranean;       |
|               |               |                |                                  |                         | Tyrrenian Sea        |
| Erik - 2015   | 21-22 May     | Mediterranean  | Shallow warm core                | (Flaounas et al.,       | Ionian Sea           |
|               | 2015          | Tropical       |                                  | 2023)                   |                      |
|               |               | Depression     |                                  |                         |                      |
| Trixie - 2016 | 28 October-1  | Mediterranean  | Deep warm core                   | (Dafis et al., 2020; Di | Ionian Sea, western  |
|               | November 2016 | hurricane      |                                  | Francesca et al.,       | Greece               |
|               |               |                |                                  | 2025; Saraceni et al.,  |                      |
|               |               |                |                                  | 2023)                   |                      |

| Caulonia -    | 16-17 March     | Mediterranean  | Shallow warm core                  | (Flaounas et al.,      |                       |
|---------------|-----------------|----------------|------------------------------------|------------------------|-----------------------|
| 2016          | 2016            | Tropical       |                                    | 2023)                  |                       |
|               |                 | Depression     |                                    |                        |                       |
| Numa – 2017   | 15-19           | Mediterranean  | Deep warm core                     | (Marra et al., 2019;   | Adriatic Sea,         |
|               | November 2017   | hurricane      |                                    | Tiesi et al., 2021)    | Greece, Albania       |
| Zorbas -      | 27-29           | Mediterranean  | 27 September 2018: deep cold core  | (de la Vara et al.,    | Southeastern          |
| 2018          | September       | hurricane      | 28 September 2018: deep            | 2021)                  | Sicily; Peloponnese   |
|               | 2018            |                | symmetric warm core                |                        | area                  |
| Vaia – 2018   | 27-29 October   | Extratropical  | Explosive cyclogenesis;            | (Davolio et al., 2020; | Ligurian coasts;      |
|               | 2018            | cyclone        |                                    | Federico et al., 2021; | North Adriatic        |
|               |                 |                |                                    | Giovannini et al.,     |                       |
|               |                 |                |                                    | 2021)                  |                       |
| Trudy - 2019  | 11-13           | Mediterranean  | Deep warm core                     | (Listowski et al.,     | Balearic Islands      |
|               | November 2019   | Tropical Storm |                                    | 2022)                  |                       |
| Ianos - 2020  | 15-20           | Mediterranean  | Deep warm core                     | (Comellas Prat et al., | Ionian Islands,       |
|               | September       | hurricane      |                                    | 2021; D'Adderio et     | western Greece        |
|               | 2020            |                |                                    | al., 2022; Jangir et   |                       |
|               |                 |                |                                    | al., 2024;             |                       |
|               |                 |                |                                    | Lagouvardos et al.,    |                       |
|               |                 |                |                                    | 2022)                  |                       |
| Apollo -      | 25 October-1    | Mediterranean  | Shallow warm core                  | (Menna et al., 2023;   | Sicily (Italy), Malta |
| 2021          | November 2021   | hurricane      |                                    | Panegrossi et al.,     |                       |
|               |                 |                |                                    | 2023)                  |                       |
| Blas - 2021   | 06-18           | Mediterranean  | Shallow warm core                  | (Mourre et al., 2023)  | Balearic Islands;     |
|               | November 2021   | hurricane      |                                    |                        | Spain, Algeria        |
| Ciprian –     | 16-20 October   | Extratropical  |                                    | (Scardino et al.,      | Cyprus Island         |
| 2022          | 2022            | cyclone        |                                    | 2024)                  |                       |
| Helios - 2023 | 8-11 February   | Mediterranean  | 8-9 February 2023: Deep cold core; | (D'Adderio et al.,     | Southeastern Sicily   |
|               | 2023            | hurricane      | 10 February 2023: warm core        | 2023)                  |                       |
| Juliette -    | 28 February – 2 | Mediterranean  |                                    | (D'Adderio et al.,     | Balearic Islands,     |
| 2023          | March 2023      | hurricane      |                                    | 2023)                  | Sardinia (Italy)      |
| Daniel - 2023 | 6-12 September  | Mediterranean  | 6 September 2023: Cold core;       | (Argüeso et al., 2024; | Greece, Turkey,       |
|               | 2023            | hurricane      | 8 September 2023: deep warm core   | Flaounas et al., 2024) | and Libya             |

## 2.2 Variations of Chl-a in offshore and nearshore areas





The analysis of Chl-a focused on the main variations observed offshore and in nearshore areas that were greatly impacted by cyclones. For this study, the nearshore was defined as the buffer zone from the coastline to the global average depth of closure of 13 m (Athanasiou et al., 2019). The coastline data were obtained from the European Environment Agency, and the depth data were extracted from the GEBCO bathymetric grid (GEBCO - The General Bathymetric Chart of the Oceans, 2020).

Offshore, the focus was on the variations in the deep chlorophyll maximum (DCM), using two kinds of datasets:

- Biogeochemistry parameters from Copernicus reanalysis products (Cossarini et al., 2021) used to assess Chl-a changes via cross-section analysis and vertical profiles across different cyclone phases (from cyclogenesis to cyclolysis)
- ARGO-Float observational data (https://dataselection.euro-argo.eu/) used to provide vertical Chl-a profiles within a variable time range (from 1 day before cyclogenesis to 1 day after cyclolysis) down to a water depth of 500 m.

An uncertainty assessment of reanalysis products was performed calculating the root mean square error (RMSE) in comparison with the observational data of the ARGO-Float (Fig. 3 and Supplementary Table S1).

The analysis of Chl-a vertical profiles was performed by measuring the difference in water depth of the DCM, in order to highlight uplift and downlift movements of the DCM.

Figure 3. Comparison between reanalysis products and ARGO float data for the medicane Zorbas; the Chl-a profiles from reanalysis were extracted in the same location as the float measurements.

Subsequently, the DCM (Deep Chlorophyll Maximum) uplift/downlift was assessed across the entire cyclone-affected area through a spatial analysis of DCM differences between pre-cyclone conditions and after the cyclone's lifetime.

Horizontal patterns of Chlorophyll-a (Chl-a) concentrations in the nearshore were derived from Sentinel-2 MSI data using the C2RCC (Case-2 Regional CoastColour) processor in ESA's SNAP software, which applies a neural-network-based atmospheric correction and inherent optical property (IOP) inversion to estimate water constituents in optically complex waters. Prior to chlorophyll retrieval, Sentinel-2 Level-1C top-of-atmosphere (TOA) radiance data were preprocessed with C2RCC, which incorporates a bio-optical model to derive water-leaving reflectances and subsequently estimate Chl-a concentrations through IOP inversion. Sentinel 2 images were selected prior to the cyclone onsets and after the cyclone impact, in order to highlight the changes in Chl-a caused by rainfall and the dynamics of surface currents.





170

## 2.3 Joint analysis of Chl-a variations and biogeochemical parameters

To better understand the mechanisms influencing Chl-a and DCM dynamics, biogeochemical parameters (such as nitrate distribution) and physical parameters (currents and MLD) were overlaid with these variables to identify upwelling/downwelling effects. Currents were assessed from Copernicus reanalysis (MEDSEA\_MULTIYEAR\_PHY\_006\_004 (Clementi et al., 2019; Escudier et al., 2020)), considering eastward (u0) and northward (v0) sea water velocities up to 500 m of the water column. These components were combined to obtain the current speed in the cross-section:

Current speed = 
$$\sqrt{u_0^2 + v_0^2}$$
 (Eq.4)

This approach highlighted how DCM dynamics respond to the cyclone passage in different locations. Additionally, a logical MLD map was created by comparing surface MLD values one day before and one day after each cyclone event, in order to correlate changes in DCM with shoaling/deepening of MLD. A spatial difference between the two MLD surfaces was performed for the entire Mediterranean basin following the relationship (Eq. 5):

$$sign\big(MLD_{dif}\big) = \begin{cases} deepening & if \ MLD_{dif} = MLD_{t2} - MLD_{t1} \geq 0 \\ shoaling & if \ MLD_{dif} = MLD_{t2} - MLD_{t1} < 0 \end{cases} (Eq.5)$$

Where:

- $sign(MLD_{dif})$  reports the sign of the difference of MLD;
- $MLD_{t2}$  indicates the MLD values at the end of cyclone lifetime;
- *MLD*<sub>t1</sub> indicates the MLD values one day before the cyclone onset.

In this way, negative values are representative of MLD shoaling, while positive values are representative of MLD deepening.

## 3. Results


## 195 3.1 Translation speed of the Mediterranean cyclones

The reanalysis products enabled assessment of key intensity parameters influencing Chl-a dynamics. Specifically, we extracted and tabulated maximum wind speed, MSLP minima, and translation speed in the appendix A.1 (**Table A.1**). The analysis of the MSLP minima revealed several slow-movement phases in the TLCs and ECs lifetime (**Supplementary Material S1**). Cyclones characterized by slow-moving phases and by a significant intensity revealed the greatest changes in the Chl-concentrations. In particular, a DCM uplift appeared for the most intense, slow-moving phases of the cyclones. For example, medicane Zorbas exhibited different phases of movement:

- A first slow-movement phase from 27 September 2018, 10:00 UTC to 27 September 2018, 17:00 UTC,
- a second, slow-moving phase from 27 September 2018, 18:00 UTC to 28 September 2018, 18:00 UTC,
- a third slow-moving phase from 29 September 2018, 00:00 UTC to 29 September 2018, 07:00 UTC.
- During the second and final quasi-stationary phase, medicane Zorbas reached its maximum intensity, characterized by low MSLP values (below 997.5 hPa). A similar behavior was observed for medicanes Ianos and Apollo (Fig. 4b-c):
  - medicane Ianos displayed very slow movement in its most intense phase, from 16 September 2020, 17:00 UTC to 19 September 2020, 05:00 UTC;
  - medicane Apollo showed an initial slow-movement phase from 28 October 2021, 14:00 UTC to 28 October 2021, 21:00 UTC, followed by another slow-moving phase lasting from 29 October 2021 to 30 October 2021, 05:00 UTC.
    Among the other cyclones considered in this study, some extratropical cyclones—Janet (November 2007), and Ciprian (October 2022)—exhibited slow movement in their most intense phases (Supplementary Material S1). Other extratropical cyclones, such as Vaia, did not experience a slow-moving phase and therefore no significant DCM uplift occurred.

Figure 4. MSLP minima displacement tracked along medicane trajectories, with graphs showing time series of central (minimum) pressure and relative displacement. Slow-moving phases (red rectangles) are illustrated for: (a) Zorbas, (b) Ianos, and (c) Apollo.

## 3.2. Heat fluxes and biogeochemical parameters




The integrated analysis of ERA5 and Copernicus reanalysis data revealed a sharp intensification (in absolute values) in downward heat fluxes (ranging from -210 to -700 W/m²) during the cyclogenesis and most intense phase of TLCs. The most extreme intensifications were observed for medicane Zorbas (extreme of -606 W/m²) and Qendresa (extreme of -655 W/m²), coinciding with their peak intensities (**Table 2**). However, medicane Qendresa showed a short quasi-stationarity phase differently from medicane Zorbas. Weaker Mediterranean storms (e.g., Akle-2011, Erik-2015) exhibited weaker heat fluxes anomalies.

Table 2. Values of heat fluxes and total heat lost within the region affected by Chl-a changes for each cyclone.

| Cyclone         | Day              | Minimum value of heat flux | Total heat lost within |
|-----------------|------------------|----------------------------|------------------------|
|                 |                  | during the cyclone         | the region affected by |
|                 |                  | occurrence (W/m²)          | Chl-a changes (10^18J) |
| Zeo - 2005      | 11 December 2005 | -386                       | -41.6                  |
| Janet - 2007    | 13 October 2007  | -255                       | -37.7                  |
| Akle - 2011     | 01 January 2011  | -139                       | -5.39                  |
| Rolf - 2011     | 05 November 2011 | -317                       | -11.6                  |
| Qendresa - 2014 | 7 November 2014  | -655                       | -26.2                  |
| Xandra - 2014   | 30 November 2014 | -260                       | -19.2                  |

| Erik - 2015     | 21 May 2015       | 102  | 10.9  |
|-----------------|-------------------|------|-------|
| Caulonia - 2016 | 16 March 2016     | 36   | 1.45  |
| Trixie - 2016   | 27 October 2016   | -269 | -45   |
| Numa - 2017     | 13 November 2017  | -300 | -64.3 |
| Zorbas - 2018   | 26 September 2018 | -606 | -46.3 |
| Vaia - 2018     | 29 October 2018   | -334 | -8.02 |
| Trudy - 2019    | 11 November 2019  | -546 | -13.1 |
| Ianos - 2020    | 14 September 2020 | -268 | -18.6 |
| Apollo - 2021   | 27 October 2021   | -210 | -45.8 |
| Blas - 2021     | 05 November 2021  | -275 | -67.9 |
| Ciprian - 2022  | 12 October 2022   | -181 | -18.5 |
| Helios - 2023   | 8 February 2023   | -498 | -13.1 |
| Juliette - 2023 | 27 October 2023   | -202 | -8.22 |

A general shoaling of biogeochemical parameters was observed concurrently with the intensification of the heat fluxes (**Supplementary Material S2**). The values of total heat lost (E<sub>total</sub>) within the region affected by the cyclone impact were obtained through the double integral of the surface downward heat flux (Eqs. 1 and 2). For the most intense cyclones, intense heat fluxes were observed, like in medicane Zorbas (-4.63E+19 J), medicane Numa (-6.43E+19 J), and medicane Blas (-6.79E+19 J). Conversely, weak cyclones were characterised by smaller heat loss, as happened for cyclones Akle (-5.39E+18 J), Erik (1.09E+19 J) and Caulonia (1.45E+18 J) (**Supplementary Material S3**). Pearson's correlation coefficient r between total energy (net heat loss) and DCM uplift resulted equal to -0.64, indicating an inverse relationship, thus indicating that a greater heat loss is associated with a greater DCM uplift.



The primary changes in these biogeochemical parameters were reflected in both the uplift of the DCM and patterns of alternating upwelling/downwelling currents (**Fig. 5**). Cross-sectional analysis further revealed that the DCM uplift was more pronounced within the cyclone center (**Fig. 5c–d**). On the outskirts of the cyclone, current vectors indicate downwelling components, which determined localized downlift in deep nutrients, like nitrate.

Figure 5. Heat fluxes and cross-section of biogeochemical parameters during medicane Zorbas (27–29 September 2018); a-b) Heat fluxes maps assessed on 27/09/2018 and 28/09/2018, with cross-sections marked in red (x in red indicates the minimum MSLP value recorded on each date); c-d) Cross-sections showing: Chlorophyll-a concentration (viridis colormap), Nitrate concentration in mmol/m³ (white contour lines), Current vectors (plasma-colored arrows), Mixed layer depth (MLD, red dashed line).

# 3.3 Uplift of the Deep Chlorophyll Maximum (DCM) in Mediterranean Offshore Regions





Vertical profiling of Chl-a from reanalysis and ARGO float data demonstrated a significant uplift of the DCM along the track of the most intense Mediterranean cyclones, particularly for those events characterized by slow-moving phases. The correlation between the number of hours of slow-moving phases and DCM uplift revealed a Pearson's r equal to 0.80, with a highly significant relationship between longer slow-moving phases and greater DCM uplift. Cyclones persisting in a slow-moving state cause ~40–60 m DCM uplift, likely due to prolonged wind-driven mixing and upwelling. Conversely, short-duration events (less than 20 hours) show minimal DCM uplift (0–10 m).

In particular, intense tropical-like cyclones, with slow-moving or quasi-stationary phases (e.g. Numa, Zorbas, Apollo, Blas, Helios), were characterized by a greater DCM uplift in their most intense phases (**Fig. 6**). In fact, a limited vertical (deep) wind shear is characteristic of the environment where medicane form and develop their warm core; such conditions favour their slow movement (Cavicchia et al., 2014). Conversely, other events, comparable in intensity but less stationary, experienced lower DCM uplift (e.g. Qendresa, Vaia, Daniel). Other events, generally of weaker intensity, were characterized by an opposite behaviour, with a DCM downlift near the cyclone center (Akle, Erik, Caulonia, Juliette).

Considering the lower resolution of the reanalysis products with respect to floats, RMSE assessments were performed to better define the significance of the DCM uplift/downlift (**Supplementary Table S1**). The RMSE values were below the

Figure 6. DCM uplift measured at the cyclone centers from the Copernicus reanalysis dataset considering as temporal range the difference between 1 day prior the cyclogenesis and 1 day after the cyclolysis (maximum total duration of 120 hours); different bars indicate different locations (blue for the cyclogenesis and light blue for most intense phase locations).

The DCM uplift was accompanied by evidence of shoaling on biogeochemical parameters. As an example, during medicane Zorbas, the DCM uplifted by 15 m (duration of 24 hours) in cyclogenesis regions and 30 m in the most intense phase (duration of 68 hours) (**Fig. 7a-b**), coinciding with intensified upwelling in the cyclone center. Cross-sectional analysis revealed that the uplift was accompanied by:

- Enhanced nitrate concentrations (up to 0.8 mmol/m³) near the euphotic zone.
- MLD shoaling in the cyclogenesis and most intense phase, with reductions greater than 5 m in regions of persistent cyclone influence.
- MLD deepening along the peripheral borders of the cyclone;





- Strong vertical currents, with upwelling velocities exceeding 0.05 m/s in the cyclone center.

Spatial analysis of DCM changes revealed widespread uplift patterns following cyclone trajectories (**Supplementary Material S4**). Medicane Zorbas exhibited particularly intense localized DCM uplift, reaching 40 m in certain areas (**Fig. 8a**); conversely, DCM downwelling was observed along the southeastern Sicily and Peloponnese coastal regions, where concurrent MLD deepening occurred (**Fig. 8b**). These MLD patterns were likely driven by a circulation associated with

downwelling in the outskirts of the cyclone during its most intense phase. Notably, Kassis and Varlas, (2021) reported MLD deepening during Zorbas, contrasting with our core-region shoaling. This discrepancy likely reflects spatial and methodological differences: their study captured basin-scale adjustments (e.g., Atlantic inflow), whereas our analysis focused on the cyclone center, where wind-driven upwelling and heat loss dominated (**Fig. 7-8**). Kassis and Varlas (2021) noted that Atlantic water intrusion could have disrupted local upwelling, leading to net MLD deepening in offshore areas of southeastern Sicily and the Pelopponese. Chl-a signal resulted to be characterized by a sustained bloom, as also highlighted in other studies (Kotta, 2023; Kotta et al., 2017; Kotta and Kitsiou, 2019b). On the other hand, the ARGO data revealed that DCM uplift persisted also after the cyclolysis in the center of the cyclone.

Figure 7. Spatial analysis of MLD changes and vertical Chl-a concentrations derived from reanalysis products (cell resolution of 0.04x0.04 degrees) during the occurrence of medicane Zorbas; a) changes in MLD assessed from 26/09/2018 to 29/09/2018, red surfaces indicate the MLD shoaling, while blue surfaces indicate the MLD deepening; b) changes in vertical profiles of Chl-a assessed in the cyclogenesis area (P1); c) changes in vertical profiles of Chl-a assessed in the proximity southeastern Sicily (P2); d) vertical profile of Chl-a assessed in the proximity of Peloponnese coasts.

Figure 8. Spatial analysis of DCM changes during the occurrence of medicane Zorbas correlated to the MLD deepening/shoaling; a) DCM depth difference assessed for the areas influenced by medicane Zorbas (difference between DCM at 29 September and DCM at 26 September 2018); b) MLD difference assessed in the same period (black line indicates the cyclone track).

# 3.4 Nearshore Chl-a Variability Driven by Runoff and Surface Currents




The development of intense TLCs and ECs was associated with an increase in Chl-a concentrations along the affected coasts, due to two different kinds of components: a redistribution of suspended organic matter caused by ocean currents and an increase in sediment-laden plumes.

In particular, cyclones characterized by the absence of heavy rainfall along the coastal areas showed an increase in Chl-a concentration in the nearby sea surface. This is the case of medicane Zorbas whose circulation impacted southeastern Sicily far from its centre. The increase of Chl-a concentration appeared to be related to the redistribution of suspended organic matter from the coast toward the sea. These dynamics seem to be related to the contributions of ocean surface currents during the cyclone impact. Sentinel-2 imagery highlighted that medicane Zorbas generated a gyre off southeastern Sicily, transporting high-Chl-a waters (>1.5 mg/m³) offshore (**Fig.9 a-b**). Conversely, areas greatly impacted by cyclone-driven rainfall highlighted plume formation several kilometers long, as observed during Daniel impacting Libya (11 September 2023). This event caused extreme surface runoff with a plume formation that determined an increase of Chl-a by 2–4 mg/m³ along Libyan coasts and over the open sea some tens of km far away (**Fig. 9 c-d**).

Figure 9. Surface changes on the Chl-a assessed from Sentinel 2 MSI images with current vectors derived from Copernicus reanalysis (background maps provided by © *Google Earth*); a) Chl-a concentration in southeastern Sicily prior to the occurrence of medicane Zorbas (27 September 2018); b) Chl-a concentration after the impact of medicane Zorbas (30 September 2018); c) Chl-a concentration along the Libyan coasts prior to the occurrence of Daniel (07 September 2023); d) Chl-a concentration after the impact of Daniel (12 September 2023).

## 4. Discussion





The analysis performed on the MSLP minima of Mediterranean cyclones revealed that slow-moving cyclones characterized by high intensity (low MSLP values and significant amount of net heat loss) could lead to a significant variation in the biogeochemical parameters. According to Piontkovski and Al-Hashmi (2014), slow-moving cyclones can drive an increase in Chl-a concentration, considering a threshold for the translation speed of 36 km/h. In our case, we considered a similar threshold of 36 km/h assessed through ERA5 MSLP hourly data. Pearson correlation revealed a positive correlation between the duration of slow-moving phases and DCM uplift (**Fig. 10a**). Prolonged durations (e.g., Ianos-2020; 79 hours) drive up to 60 m of DCM uplift, while short-lived stationary phases (e.g., Rolf-2011; 9 hours) show minimal effect. Additionally, a prolonged persistence of strong cyclones can determine heavy rainfall in a given location, favoring vertical mixing in seawater and increasing phytoplankton blooms (Owen et al., 2021; Vargas-Yáñez et al., 2022), and also influencing the productivity during the spring season (Li et al., 2024a).

Air-sea heat exchange may also play an important role in generating eddy-induced upwelling, particularly in driving sea surface temperature (SST) cooling (Liu et al., 2020; Varlas et al., 2020). Evaluation of the surface heat fluxes allowed to obtain the E<sub>total</sub> lost within the sea regions affected by the cyclones. Negative values of E<sub>total</sub> indicate heat energy lost by the sea surface, which are related to air-sea interaction processes during the cyclone passage (Liu et al., 2019; Walker et al.,

2005). In particular, a negative correlation was observed between E<sub>total</sub> lost by sea water during the cyclone occurrence and DCM uplift (Fig. 10b). This pattern indicates that DCM uplift is strongly influenced by cooling processes occurring during intense cyclones, such as upwelling, radiative heat loss, and vertical mixing. These processes may be enhanced in the case of intense cyclones that persist in a given area for longer. However, Fig. 10c shows that, among slow-moving cyclones, the DCM uplift increases weakly with the translational speed; conversely, Figure 10d shows that a longer lifetime is associated with a greater DCM uplift.

Figure 10 – Correlation among DCM uplift with duration and intensity of the Mediterranean cyclones; a) positive correlation between duration of slow-moving phases and DCM uplift; b) negative correlation between total energy lost by sea water and DCM uplift; c) correlation between translation speed assessed during slow-moving phase and DCM uplift; d) positive correlation between total duration of cyclones (including no-stationary phases) and DCM uplift.


A decrease in the intensity of heat surface fluxes may be due to the sea surface cooling (Avolio et al., 2024; Kotta, 2023; Kotta and Kitsiou, 2019b; Scardino et al., 2024) and the consequent cyclone weakening in areas that are already influenced


Figure 11. Decrease in heat fluxes and sea surface temperature (SST) during the occurrence of medicane Zorbas; a) day of highest intensity of heat fluxes; b) time series of SST and surface heat fluxes extracted in the cyclogenesis location (0.04 x 0.04 degrees grid resolution).

Slow-moving cyclones can enhance wind-driven upwelling through the wind pump effect (Mourre et al., 2023). In fact, according to Yang et al. (2024), while TLCs are responsible for a pronounced sea surface cooling along their track, the translation motion of a TLC could generate an inertial pumping in its lee, leading to a pattern of alternating downwelling and upwelling areas (Geisler, 1970; Shay et al., 1992; Suzuki et al., 2011; Yang et al., 2024), mainly in the cyclone peripheries (Yang et al., 2024).

Owing to the spatial distribution of wind stress, downwelling outside the cyclone's maximum radius is generally weaker than upwelling near the cyclone's center (Fig. 12a). Therefore, a significant DCM uplift is observed near the cyclone center related to strong upwelling currents, while the periphery and rainbands are characterized by an alternating pattern of downwelling and upwelling and weak DCM changes (Fig. 12b). Notably, when TLCs translate more rapidly, the upper

Figure 12. Features of Chl-a changes related to the structure of the medicane Zorbas; the cyclone center highlighted the DCM uplift and upwelling, while periphery and rainbands display alternating downwelling and upwelling; a) satellite image of medicane Zorbas (MODIS image-acquired on 28 September 2018) highlighting the cyclone center and periphery; b) cross-section of biogeochemical parameters referred to the time of MODIS image showing: Chlorophyll-a concentration (viridis colormap),




A different behavior was observed in the nearshore areas through the analysis of multispectral satellite images. The contribution from suspended organic matter increased the Chl-a concentrations in the nearshore, enhanced by extreme rainfall. This occurred in response to the strong surface runoff influencing the coastal areas. However, some Mediterranean hurricanes did not experience extreme rainfall in the impacted areas, as during medicane Zorbas in southeastern Sicily (Kushabaha et al., 2024; Scicchitano et al., 2021). The high Chl-a concentration observed nearshore via satellite on 27 September 2018 contributed to elevated Chl-a levels in offshore areas following the medicane's impact. A joint analysis of satellite-derived Chl-a concentrations (30 September 2018, one day after medicane Zorbas) and surface current distribution revealed a significant increase in subsurface seawater Chl-a. This enhancement was driven by a gyre system that transported suspended organic matter seaward (Fig. 13).

Figure 13. Dynamics of Chl-a on the offshore and nearshore area during the impact of medicane Zorbas in southeastern Sicily; a) areal Chl-a concentration assessed through Sentinel 2 image after the impact of medicane Zorbas (background maps provided by © Google Earth), red line indicates the cross-section in b); b) cross-section of biogeochemical parameters extracted from Copernicus reanalysis products; c) vertical profiles of Chl-a changes extracted in P1; d) vertical profiles of Chl-a changes extracted in P2.

Our analysis of twenty Mediterranean cyclones reveals that the pre-storm oceanographic condition, particularly the MLD and the strength of the seasonal thermocline, is fundamental for biogeochemical response. The general pattern that emerges from a composite analysis is one of seasonal control (Menkes et al., 2016). Strong stratification during summer and early autumn typically inhibited vertical mixing, limiting nutrient uplift (D'Ortenzio and Ribera d'Alcalà, 2009; Teruzzi et al., 2021). However, slow-moving cyclones may overcome this barrier through prolonged wind-driven upwelling and turbulent mixing, leading to significant DCM uplift (40–60 m) and Chl-a concentration increases (Jangir et al., 2024; Kotta and Kitsiou, 2019b). The prolonged wind forcing and significant net heat loss of these cyclones provide the sustained energy required to erode the stratification, shoal the MLD, and drive significant upwelling (Lin, 2012; Menkes et al., 2016; Walker et al., 2005).




In contrast, weak stratification during late autumn and winter facilitated easier mixing, but the response was less pronounced when the pre-storm MLD was already deep or nitrate levels were low, as observed during medicane Helios (D'Adderio et al., 2023). MLD shoaling was most pronounced in areas with initially shallow or seasonally variable MLD (Ricchi et al., 2019, 2020; Vargas-Yáñez et al., 2022). For instance, medicane Apollo (October 2021) induced a 25 m MLD shoaling in the central Mediterranean, where summer stratification was transitioning (Menna et al., 2023). Two primary factors characterized the Chl-a changes in the Mediterranean basin: i) the nutrient limitation due to the pre-cyclone mixing, which may have already homogenized the upper water column, depleting subsurface nutrient reservoirs; ii) dilution effect due to a deep MLD, with distribution of uplifted nutrients over a larger volume.

The most substantial Chl-a increases occurred in regions with abundant subsurface nitrate reservoirs, like in the Ionian Sea (Lazzari et al., 2016)), particularly during slow-moving phases of medicanes like Zorbas, Apollo, and Ianos (Jangir et al., 2024; Kotta and Kitsiou, 2019b). By way of example, medicane Zorbas produced a strong biogeochemical response in the Ionian Sea, where moderate pre-storm stratification and high subsurface nitrate concentrations prevailed, but weaker effects occurred in the Peloponnese due to downwelling-favorable currents (Kassis & Varlas, 2021). Sentinel imagery also revealed an offshore transport of suspended organic matter and Chl-a, evident in southeastern Sicily during Medicane Zorbas (Fig. 13). Conversely, cyclone Ciprian triggered coastal upwelling near Cyprus (Fig. 14), where nitrate-rich waters enhanced the Chl-a response. In contrast, weaker or faster-moving cyclones (e.g., Erik-2015) had minimal effects in highly stratified or oligotrophic areas, demonstrating how pre-cyclone ocean conditions, combined with cyclone evolution, critically shape biogeochemical responses (Macías et al., 2014; Mélin et al., 2017; Menkes et al., 2016).

Conversely, other cyclones, responsible for intense rainfall, triggered intense surface runoff, which was reflected in high Chl-a concentrations resembling plumes near river mouths, as observed off Libya coast following the impact of medicane Daniel in 2023 (Fig. 9d). Furthermore, a correlation between coastal upwelling and Chl-a increase was also observed along the Balearic Islands during the impact of Blas (Mourre et al., 2023) and along the Cyprus coasts during the impact of Ciprian (analysis reported in Fig. 14).

Figure 14. Coastal upwelling induced DCM uplift and increased Chl-a concentration during the EC Ciprian (16-20 October 2022) (background maps provided by © *Google Earth*); a) Chl-a assessment from Sentinel 2 images acquired on 15/10/2022 (prior of cyclone onset); b) Chl-a assessment from Sentinel 2 image acquired on 20/10/2022 (end of the cyclone lifetime); c) cross-section of biogechemical parameters revealing a coastal upwelling on the western side of Cyprus island.

Other authors reported a different behavior for other kinds of intense weather systems worldwide (Chen et al., 2017; Macías et al., 2014; Wu et al., 2007). For example, Typhoon Trami in the northwest Pacific Ocean triggered an increase of Chl-a

concentration in the surface layer related to MLD deepening, and to the consequent enrichment of the surface layer with DCM waters through mixing (Chai et al., 2021). Mediterranean cyclones induce DCM uplift through wind-driven mixing and upwelling, bringing nutrients to the euphotic zone. This aligns with observations in the Pacific and Atlantic, where cyclones enhance primary productivity via vertical mixing (Lin, 2012; Zhao et al., 2008). The dome-shaped uplift of biogeochemical parameters (e.g., MLD shoaling, nitrate enrichment) near the cyclone center mirrors patterns seen in tropical cyclones (Price, 1981; Sanford et al., 1987; Walker et al., 2005). However, Mediterranean cyclones (especially medicanes) are smaller and shorter-lived than tropical cyclones, yet their slow movement (e.g., Zorbas: 11.5 km/h) prolongs mixing, creating localized but intense biogeochemical responses. Conversely, tropical cyclones typically cover larger areas but with faster translation speeds (e.g., 20-30 km/h; Fu et al., 2014)). Mediterranean coastal upwelling (e.g., during Blas or Ciprian) is often wind-driven and localized, unlike large-scale equatorial upwelling zones affected by tropical cyclones. However, the MLD in the Mediterranean Sea is highly influenced by seasonal variations and can differ significantly between summer and winter (Barboni et al., 2023; D'Ortenzio et al., 2005). The MLD depth can influence the heat fluxes and can enhance convection during TLCs (Ricchi et al., 2019, 2020). In our study, we observed MLD shoaling in areas where cyclones exhibited slow or quasi-stationary movement, while the peripheries of the cyclones were characterized by MLD deepening. This pattern can be attributed to strong upwelling in the cyclone eye (inducing MLD shoaling) and alternating downwelling and upwelling in the peripheral regions (promoting MLD deepening). The events characterized by strong vertical shear due to the water currents revealed a water mixing that enhances the MLD shoaling, as observed during medicane Trixie and cyclone Trudy (see cross-sections in the Supplementary Material S2). As a result, the MLD may exhibit a dome-shaped structure along the tracks, as already reported for medicane Apollo (Menna et al., 2023).








The confined nature and complex coastline of the Mediterranean Sea amplify the coastal impacts of TLCs. Our analysis reveals that excluding cyclones within 100 km of the coast removes the most Chl-a concentrations (>2 mg/m³), which are exclusively associated with terrestrial processes like runoff-driven plumes (e.g., medicane Daniel (Normand and Heggy, 2024)) or the coastal resuspension and advection of organic matter (e.g., medicane Zorbas; (Kotta, 2023)). In contrast, the purely offshore response, though covering a larger area, yields more modest Chl-a increases (0.01–0.1 mg/m³) driven solely by DCM uplift. This highlights that the most dramatic surface blooms are a direct product of the Mediterranean's unique biogeography (Macías et al., 2014; Marañón et al., 2021), while the subsurface offshore uplift remains a critical mechanism for enhancing nutrient cycling in the basin's oligotrophic open waters(D'Ortenzio and Ribera d'Alcalà, 2009)

Furthermore, the Mediterranean Sea is nutrient-limited, with surface Chl-a typically ranging from 0.05 to 0.3 mg/m³ (Marty and Chiavérini, 2002; Moutzouris-Sidiris and Topouzelis, 2021; Teruzzi et al., 2021). Cyclone occurrence could disrupt the water column stratification, injecting nitrate (0.5–0.8 mmol/m³) and phosphate into the euphotic zone (**Fig. 5, 7**). This pulsed nutrient supply can be particularly significant in late summer and autumn, when the Mediterranean stratified surface waters are most nutrient-depleted (D'Ortenzio et al., 2005; D'Ortenzio and Ribera d'Alcalà, 2009). For instance, slow-moving medicanes, like Ianos (September 2020), drove 30–60 m uplifts of DCM, enhancing Chl-a concentrations by 0.01–0.02 mg/m³—comparable to the effects of seasonal riverine inputs in coastal zones (Ludwig et al., 2009).

These nutrient pulses can shift plankton community composition, favoring larger, fast-growing diatoms over smaller picoplankton that dominate under oligotrophic conditions (Marty and Chiavérini, 2002). Such shifts have been observed following medicane Apollo (2021), where diatom blooms coincided with DCM shoaling (Menna et al., 2023). This transition from microbial loop-dominated recycling to a classical diatom-copepod food web can enhance carbon export efficiency (Chen et al., 2017; Cullen, 2015), potentially offsetting productivity declines associated with marine heatwaves (Androulidakis and Pytharoulis, 2025; Darmaraki et al., 2019; Izquierdo et al., 2022; Li et al., 2024a).

The cumulative impact of cyclones on annual productivity budgets is likely modest ( $\leq 5-10\%$  in affected areas), but their role in late-season nutrient injection and carbon export could grow under climate change, especially if warming SSTs intensify cyclone-driven mixing (Avolio et al., 2024; González-Alemán et al., 2019). Future studies should quantify how these event-scale perturbations integrate into broader biogeochemical cycles and whether they enhance ecosystem resilience to stratification.

#### 5. Conclusions



- This study elucidates the biogeochemical responses of the Mediterranean Sea to TLCs and ECs, with a focus on Chl-a dynamics. Our findings highlight that slow-moving cyclones induced pronounced DCM uplift offshore, with values up to 40 to 60 m. This uplift is driven by a combination of intense heat loss at the sea surface, wind-driven water mixing, intensified upwelling currents, and shoaling of the MLD, which collectively enhance nutrient availability in the euphotic zone.
- The analysis revealed distinct spatial patterns in Chl-a variability. Offshore regions along the tracks, particularly within cyclone centers, experienced significant DCM uplift due to upwelling, while peripheral areas showed alternating upwelling/downwelling currents with minimal DCM changes. This behaviour is reflected in a dome-shaped DCM along the cyclone track. Nearshore regions, influenced by surface runoff and suspended organic matter, displayed elevated Chl-a concentrations (up to 10 mg/m³), often forming plumes near river mouths. These nearshore effects were further modulated by cyclone-induced rainfall and surface current dynamics.
- Our results underscore the dual role of Mediterranean cyclones in shaping marine productivity: offshore, through physical mechanisms that redistribute nutrients vertically, and nearshore, through terrestrial inputs and surface transport. These insights have implications for predicting primary productivity shifts under changing cyclone regimes, particularly in the context of climate change. In this perspective, the role of cyclones in late-summer and autumn biogeochemical processes should also be investigated in the light of the inhibition of spring phytoplankton blooms by increasingly frequent and intense marine heatwaves. Future research should explore the long-term ecological impacts of these episodic events and their potential feedback on Mediterranean marine ecosystems.

#### Appendix A.1

Table A.1. Intensity parameters of the analyzed Mediterranean cyclones derived from ERA5 reanalysis data, the translation speed is referred to average speed during the slow-movement phases.

| Cyclone       | Total    | Maximum | Minima | Time range of slow- | <b>Duration of</b> | Translation |
|---------------|----------|---------|--------|---------------------|--------------------|-------------|
|               | duration | Wind    | MSLP   | movement phase      | slow-              | speed       |
|               | (hours)  | Speed   | (hPa=  |                     | movement           | (km/h)      |
|               |          | (km/h)  |        |                     | phase              |             |
|               |          |         |        |                     | (hours)            |             |
| Zeo - 2005    | 72       | 93.5    | 989    | 2005-12-13 17:00:00 | 4                  | 22.24       |
|               |          |         |        | 2005-12-13 21:00:00 |                    |             |
| Janet - 2007  | 160      | 60      | 1012   | 2007-10-16 15:00:00 | 54                 | 13.1        |
|               |          |         |        | 2007-10-18 21:00:00 |                    |             |
| Rolf - 2011   | 52       | 71      | 996    | 2011-11-06 05:00:00 | 9                  | 13.25       |
|               |          |         |        | / 2011-11-06        |                    |             |
|               |          |         |        | 14:00:00            |                    |             |
| Akle - 2011   | 52       | 60      | 1009   | 2011-01-02 10:00:00 | 12                 | 8.23        |
|               |          |         |        | / 2011-01-02        |                    |             |
|               |          |         |        | 22:00:00            |                    |             |
| Qendresa -    | 54       | 87.6    | 993    | 2014-11-07 18:00:00 | 16                 | 16.88       |
| 2014          |          |         |        | 2014-11-08 10:00:00 |                    |             |
| Xandra -      | 72       | 76      | 990    | 2014-12-01 04:00:00 | 9                  | 9.68        |
| 2014          |          |         |        | 2014-12-01 13:00:00 |                    |             |
| Erik - 2015   | 30       | 55      | 1003   | 2015-05-22 00:00:00 | 6                  | 3.27        |
|               |          |         |        | 2015-05-22 06:00:00 |                    |             |
| Caulonia -    | 27       | 59      | 1011   | 2016-03-16 00:00:00 | 13                 | 25.1        |
| 2016          |          |         |        | 2016-03-16 13:00:00 |                    |             |
| Trixie - 2016 | 98       | 70      | 1012   | 2016-10-28 16:00:00 | 41                 | 20.23       |
|               |          |         |        | 2016-10-30 09:00:00 |                    |             |
| Numa - 2017   | 96       | 66      | 1004   | 2017-11-16 14:00:00 | 63                 | 13.38       |
|               |          |         |        | 2017-11-19 05:00:00 |                    |             |
| Zorbas - 2018 | 80       | 88      | 995    | 2018-09-27 06:00:00 | 68.0               | 11.49       |

|                 |     |     |        | 2018-09-30 12:00:00 |      |       |
|-----------------|-----|-----|--------|---------------------|------|-------|
| Vaia - 2018     | 48  | 100 | 994    | 2018-10-28 05:00:00 | 4    | 15.16 |
|                 |     |     |        | / 2018-10-28        |      |       |
|                 |     |     |        | 09:00:00            |      |       |
| Trudy - 2019    | 36  | 78  | 998    | 2019-11-11 00:00:00 | 21   | 14.7  |
|                 |     |     |        | / 2019-11-11        |      |       |
|                 |     |     |        | 21:00:00            |      |       |
| Ianos - 2020    | 120 | 64  | 1004   | 2020-09-16 11:00:00 | 79   | 16.85 |
|                 |     |     |        | 2020-09-19 18:00:00 |      |       |
| Apollo - 2021   | 144 | 66  | 1003   | 2021-10-28 23:00:00 | 57   | 12.89 |
|                 |     |     |        | / 2021-10-31        |      |       |
|                 |     |     |        | 12:00:00            |      |       |
| Blas - 2021     | 144 | 75  | 1005   | 2021-11-10 23:00:00 | 40   | 15.14 |
|                 |     |     |        | / 2021-11-12        |      |       |
|                 |     |     |        | 15:00:00            |      |       |
| Ciprian -       | 100 | 58  | 1005   | 2022-10-19 05:00:00 | 13   | 14.78 |
| 2022            |     |     |        | / 2022-10-19        |      |       |
|                 |     |     |        | 18:00:00            |      |       |
| Helios - 2023   | 72  | 74  | 1011   | 2023-02-09 23:00:00 | 25.0 | 21.48 |
|                 |     |     |        | / 2023-02-11        |      |       |
|                 |     |     |        | 00:00:00            |      |       |
| Juliette - 2023 | 54  | 80  | 997    | 2023-02-28 00:00:00 | 12   | 15.75 |
|                 |     |     |        | / 2023-02-28        |      |       |
|                 |     |     |        | 12:00:00            |      |       |
| Daniel - 2023   | 144 | 65  | 997.16 | 2023-09-10 11:00:00 | 18   | 17.34 |
|                 |     |     |        | 2023-09-11 05:00:00 |      |       |

**Data availability:** The dataset generated in this study, including DCM uplift and cyclone total energy, is available at: https://doi.org/10.5281/zenodo.15912789.

- Third-party repository for MSLP: ERA5: Fifth generation of ECMWF atmospheric reanalyses of the global climate.

  515 Copernicus Climate Change Service Climate Data Store (CDS). Retrieved 01/06/2025 from https://cds.climate.copernicus.eu/cdsapp#!/home
- Third-party repository for biogeochemical parameters: Cossarini, G., Feudale, L., Teruzzi, A., Bolzon, G., Coidessa, G., Solidoro, C., Di Biagio, V., Amadio, C., Lazzari, P., Brosich, A., and Salon, S.: High-Resolution Reanalysis of the Mediterranean Sea Biogeochemistry (1999–2019), Front. Mar. Sci., 8, https://doi.org/10.3389/fmars.2021.741486, 2021.
- Third-party repository for currents and MLD: 1) Escudier, R., Clementi, E., Omar, M., Cipollone, A., Pistoia, J., Aydogdu, A., Drudi, M., Grandi, A., Lyubartsev, V., Lecci, R., Cretí, S., Masina, S., Coppini, G., and Pinardi, N.: Mediterranean Sea Physical Reanalysis (CMEMS MED-Currents) (Version 1) Data set., Copernicus Monitoring Environment Marine Service (CMEMS)., https://doi.org/10.25423/CMCC/MEDSEA\_MULTIYEAR\_PHY\_006\_004\_E3R1, 2020; 2) Clementi, E., Pistoia, Escudier, R., Delrosso, D., Drudi, M., Grandi, A., Lecci, R., Cretí, S., Ciliberti, S., Coppini, G., Masina, S., and Pinardi, N.: Mediterranean Sea Physical Analysis and Forecast (CMEMS MED-Currents 2016-2019), Copernicus Monitoring Environment Marine Service (CMEMS), Dataset 1, 2, https://doi.org/10.25423/cmcc/medsea\_analysis\_forecast\_phy\_006\_013\_eas4, 2019.
- Third-party repository for direct observation: Argo. (2023). Argo float data and metadata from the Global Data Assembly Centre (GDAC). SEANOE. https://doi.org/10.17882/42182

#### **Author contribution:**

G.S, and A.K. wrote the main text, analysed climate datasets and processed the Chl-a concetration from reanalysis and remote sensing data. M.M.M., and D.B. revised the concept related to the slow-moving phases and intensity of Mediterranean cyclones and provided a revised version of the manuscript. G.S., M.M.M., and G.Scicchitano. reported a validation on the model outputs. G.Scicchitano performed the supervision and investigation and provided the resources for the research.

# Acknowledgments

540

550

This work has greatly benefited from contributions of the ERC SEED UNIBA project "Get aHead Of the MEdicanes: strategies for the COASTal environment—HOME-COAST" (Principal Investigator Giovanni Scardino, PhD). The validation methods for the Sentinel-3 dataset were supported by the Sentinel-3 Validation Team (S3VT) under the project "Frozen Coasts under Climate Change" (Principal Investigator Giovanni Scardino, PhD, ESA project PP0092792). This paper and related research have been conducted within the framework of the Italian inter-university PhD course in sustainable development and climate change (link: www.phd-sdc.it) 'Medichange' (responsible Prof. Giovanni Scicchitano).

Competing interests: The authors declare that they have no conflict of interest.

# **Funding sources**

This research has been funded by the PRIN 2022 PNRR project titled "ARCHIMEDE—Multidisciplinary approach to better define vulnerability and hazard of Medicanes along the Ionian coasts of Sicily" (CUP H53D23011380001, Principal Investigator Prof. G. Scicchitano).

555

Correspondence and requests for materials should be addressed to G.S.

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
