# Peer review of "When Storms Stir the Mediterranean Depths: Chlorophyll-a Response to Mediterranean Cyclones"

_EGUsphere, 2025_

## Author Comment (AC2)

**ZEO – December 2005**

Zeo Cyclone Track with Wind Influence Areas

Cyclone Center Pressure and Movement

**JANET – October 2007**

Janet Cyclone Track with Wind Influence Areas

[Figure]

**Cyclone Center Pressure and Movement**

**ROLF – November 2011**

**Rolf Cyclone Track with Wind Influence Areas**

[Figure]

**Cyclone Center Pressure and Movement**

**AKLE – January 2011**

**Akle Cyclone Track with Wind Influence Areas**

[Figure]

**QENDRESA – November 2014**

Qendresa Cyclone Track with Wind Influence Areas

[Figure]

**Cyclone Center Pressure and Movement**

**XANDRA – December 2014**

Xandra Cyclone Track with Wind Influence Areas

[Figure]

Gray shading: Wind influence area
(Darker = larger radius)

— Cyclone Track
● Cyclone Center

**Cyclone Center Pressure and Movement**

[Figure]

**ERIK – May 2015**

Erik Cyclone Track with Wind Influence Areas

[Figure]

**Cyclone Center Pressure and Movement**

[Figure]

**CAULONIA – March 2016**

**Caulonia Cyclone Track with Wind Influence Areas**

[Figure]

**Cyclone Center Pressure and Movement**

[Figure]

**TRIXIE – October 2016**

**Trixie Cyclone Track with Wind Influence Areas**

Gray shading: Wind influence area
(Darker = larger radius)

Cyclone Track
Cyclone Center

**Cyclone Center Pressure and Movement**

**NUMA – November 2017**

[Figure]

Numa Cyclone Track with Wind Influence Areas

Cyclone Center Pressure and Movement

**ZORBAS – September 2018**

Zorbas Cyclone Track with Wind Influence Areas

[Figure]

**Cyclone Center Pressure and Movement**

[Figure]

**VAIA – October 2018**

[Figure]

Vaia Cyclone Track with Wind Influence Areas

**Cyclone Center Pressure and Movement**

**TRUDY – November 2019**

**Trudy Cyclone Track with Wind Influence Areas**

[Figure]

Gray shading: Wind influence area
(Darker = larger radius)

— Cyclone Track
● Cyclone Center

**Cyclone Center Pressure and Movement**

[Figure]

**IANOS – September 2020**

Ianos Cyclone Track with Wind Influence Areas

[Figure]

**APOLLO – October 2021**

**Apollo Cyclone Track with Wind Influence Areas**

**Cyclone Center Pressure and Movement**

**BLAS – November 2021**

Blas Cyclone Track with Wind Influence Areas

[Figure]

Cyclone Center Pressure and Movement

**CIPRIAN – October 2022**

[Figure]

Ciprian Cyclone Track with Wind Influence Areas

Gray shading: Wind influence area
(Darker = larger radius)

Cyclone Track
Cyclone Center

Cyclone Center Pressure and Movement

**HELIOS – February 2023**

Helios Cyclone Track with Wind Influence Areas

[Figure]

Cyclone Center Pressure and Movement

**JULIETTE – February-March 2023**

[Figure]

**DANIEL – September 2023**

Daniel Cyclone Track with Wind Influence Areas

[Figure]

Cyclone Center Pressure and Movement

---

## Author Response (AR1)

Here, we provided point-by-point responses to the comments of Reviewers 1-2. Author responses are marked in red.

**Round 1 – Response to Reviewer 1**

This study presents a valuable investigation into the impact of Mediterranean cyclones on biogeochemical dynamics, specifically Chl-a. The integrated multi-platform approach (reanalysis, ARGO, Sentinel-2) enables a reasonable examination of both surface and subsurface responses. The core finding is DCM uplift driven by slow-moving cyclones and its governing physical mechanisms, which has been well studied in many other ocean basins, though it is important for understanding Mediterranean productivity. The manuscript is generally well-focused, but requires substantial review from former studies to clarify the creativity.

**Answer.** Many thanks for your valuable comments that allowed us to improve the manuscript. Here, we reported a step-by-step response to your comments, and we have highlighted the associated changes in the main text.

**Point 1.** The most important issue is the fundamental dynamics, like the cyclone-induced dynamics and pre-cyclone condition. This topic has been widely discussed and investigated in former studies, but the current study was not well reviewed for these mechanisms. It is important to incorporate the findings from former studies and highlight the creativity of the current work.

**Answer 1.** We revised the manuscript considering former studies, mainly those considered for cyclone-induced dynamics and pre-cyclone condition. In particular, a discussion for DCM and MLD depths prior the cyclone onset was inserted, also considering the season for the different cyclones (lines 408-416).

In contrast, weak stratification during late autumn and winter facilitated easier mixing, but the response was less pronounced when the pre-storm MLD was already deep or nitrate levels were low, as observed during medicane Helios(D'Adderio et al., 2023). MLD shoaling was most pronounced in areas with initially shallow or seasonally variable MLD (Ricchi et al., 2019, 2020; Vargas-Yáñez et al., 2022). For instance, medicane Apollo (October 2021) induced a 25 m MLD shoaling in the central Mediterranean, where summer stratification was transitioning (Menna et al., 2023).

The most substantial Chl-a increases occurred in regions with abundant subsurface nitrate reservoirs, like in the Ionian Sea (Lazzari et al., 2016)), particularly during slow-moving medicanes like Zorbas, Apollo, and Ianos (Jangir et al., 2024; Kotta and Kitsiou, 2019b). By way of example, medicane Zorbas produced a strong biogeochemical response in the Ionian Sea, where moderate pre-storm stratification and high subsurface nitrate concentrations prevailed, but weaker effects occurred in the Peloponnese due to downwelling-favorable currents (Kassis & Varlas, 2021). Conversely, cyclone Ciprian triggered coastal upwelling near Cyprus (Fig. 14), where nitrate-rich waters enhanced the Chl-a response . In contrast, weaker or faster-moving cyclones (e.g., Erik-2015) had minimal effects in highly stratified or oligotrophic areas, demonstrating how pre-cyclone ocean conditions, combined with cyclone evolution, critically shape biogeochemical responses (Macías et al., 2014; Mélin et al., 2017; Menkes et al., 2016)."

**Point 2.** Clearly define the criteria used to classify cyclones as "Tropical-like" (TLCs) vs. "Extratropical" (ETCs) in this specific study context. Were hybrid types included? Referencing a standard classification scheme would be helpful.

**Answer 2.** The classification for each cyclone, in terms of TLCs vs ETCs, derives from literature works (see Table 1). Hence, we did not apply a criterion to differentiate them, but we only considered the oceanographic parameters of the cyclones that influenced the Chl-a dynamics. Note that the definition of medicanes has been provided recently (Miglietta et al., 2025), which includes hybrid cyclones.

**Point 3.** Justify the selection criteria for the 20 cyclones. What thresholds (e.g., intensity, duration, size) were applied? Were specific seasons or years favored? A table listing the cyclones, their type, key characteristics (max wind, min pressure, translation speed), and impacted regions would significantly enhance transparency and reproducibility.

**Answer 3.** To justify the criteria for the 20 cyclones, we considered the strongest medicanes and some of the most intense extratropical cyclones already studied and discussed in literature (see Table 1). We inserted another table (new Table A1) in the appendix A1 that reports the key characteristics of the cyclones assessed from ERA5 (e.g., max wind, MSLP minima, translation speed). Furthermore, the impacted areas are shown in Supplementary Material S1, with a gray-shaded radius indicating the region influenced by wind stress.

**Point 4.** Provide explicit, quantitative definitions for "offshore" and "nearshore" zones used in the analysis (e.g., distance from coast, bathymetry thresholds). This is crucial for interpreting the comparative results.

**Answer 4.** A quantitative definition was inserted in subsection 2.2 (line 138). In this study, the nearshore was defined as the buffer zone from the coastline to the global average depth of closure of 13 m (Athanasiou et al., 2019). The coastline data were obtained from the European Environment Agency, and the depth data were extracted from the GEBCO bathymetric grid (GEBCO - The General Bathymetric Chart of the Oceans, 2020).

**Point 5.** The abstract highlights comparing offshore and nearshore responses, but the summary focuses primarily on DCM uplift (typically an offshore phenomenon). Elaborate on the key differences or similarities observed between these zones regarding Chl-a response (surface and subsurface), nitrate supply, and MLD changes. Was nearshore response dominated by different processes (e.g., sediment resuspension, terrestrial input)?

Answer 5. As discussed in Section 3.4, nearshore dynamics are primarily driven by runoff and surface currents. While upwelling also modulates Chl-a variability in this zone, satellite imagery clearly underscores the significant role of sediment resuspension and terrestrial inputs. This is evident in Figs. 9c–d, which shows a prominent plume generated by terrestrial discharge. Although terrestrial inputs appeared to be relevant in the nearshore, their influence has been reflected in the offshore too, in which the surface currents drove the transportation of floating sediments for several kilometers seaward.

**Point 6.** While wind-driven upwelling and air-sea heat exchange are identified as critical, can you quantify their relative contributions to the DCM uplift and resultant Chl-a changes (e.g., using correlation analysis, idealized model runs, or budget analysis within the reanalysis/ARGO data)?

**Answer 6.** A revised correlation analysis was reported in the Fig. 10, where the correlation with DCM uplift was calculated with the duration of slow-moving phases, the Total Energy due to heat fluxes, and the total duration of the events. The main results of correlation analysis are discussed in section 4, where a positive correlation was observed between slow-moving phases and DCM uplift (Fig.10a), while a negative correlation was observed between total energy of heat flux and DCM

uplift (Fig.10b). The positive correlation reflects a prolonged persistence of strong cyclones that can determine persistent upwelling, heavy rainfall in a given location, favoring vertical mixing in seawater and increasing phytoplankton blooms. Conversely, the negative correlation is related to cooling processes occurring during the intense phase of cyclones, such as upwelling, radiative heat loss, and vertical mixing.

**Point 7.** Discuss the potential influence of pre-storm oceanographic conditions (stratification strength, initial DCM depth/nitrate content, background MLD) on the magnitude of the observed response. Was the response consistent, or did it depend heavily on the pre-cyclone state?

**Answer 7**. A discussion concerning pre-cyclone conditions was inserted from line 396 to line 407:

"The concentration of Chl-a and DCM responses to Mediterranean cyclones was strongly modulated by pre-storm oceanographic conditions of the Mediterranean basin. Strong stratification during summer and early autumn typically inhibited vertical mixing, limiting nutrient uplift (D'Ortenzio & Ribera d'Alcalà, 2009). However, slow-moving cyclones (e.g., Zorbas, Ianos) overcame this barrier through prolonged wind-driven upwelling and turbulent mixing, leading to significant DCM uplift (40–60 m) and Chl-a increases (Jangir et al., 2024). In contrast, weak stratification during late autumn and winter facilitated easier mixing, but the response was less pronounced when the pre-storm MLD was already deep or nitrate levels were low, as observed during medicane Helios (February 2023; D'Adderio et al., 2023). MLD shoaling was most pronounced in areas with initially shallow or seasonally variable MLD. For instance, medicane Apollo (October 2021) induced a 25 m MLD shoaling in the central Mediterranean, where summer stratification was transitioning (Menna et al., 2023)".

**Point 8**. Detail the typical temporal lag between cyclone passage, MLD deepening, DCM uplift, and the observable surface Chl-a response. Sentinel-2 coverage is weather-limited; how did this impact capturing the peak response? Did ARGO floats provide insight into the evolution?

**Answer 8.** A new sentence was inserted in the section 2 (line 100):

"To capture the full temporal evolution of biogeochemical parameters during the cyclone passage, we analyzed Sentinel-2 and Argo data alongside ERA5 (hourly data) and CMEMS (daily data) reanalysis products. This approach provided a comprehensive overview of the event's progression, particularly revealing time-series variations in mixed layer depth (MLD) and deep chlorophyll maximum (DCM) dynamics".

Sentinel-2 coverage is weather-limited, and ARGO floats provide point data on the water vertical profile. So, the integration with reanalysis dataset allowed us to obtain a full temporal evolution of the cyclone passage. However, the outputs are provided as daily data, so quantifying the time lag between cyclone passage and DCM uplift (from the CMEMS reanalysis) is not possible. This is especially true given the intensity parameters and slow-moving phases assessed from the ERA5 hourly data.

**Point 9.** Discuss the persistence of the Chl-a signal. Was the uplift transient, or did it lead to a sustained bloom?

**Answer 9.** Chl-a signal resulted to be characterized by a sustained bloom, as also highlighted in other studies (Kotta and Kitsiou, 2009; Kotta et al. 2017). On the other hand, the ARGO data revealed that DCM uplift persisted also after the cyclolysis.

**Point 10.** Acknowledge the limitations of Sentinel-2 for Chl-a retrieval in the Mediterranean, particularly in offshore oligotrophic waters where algorithms are less robust and in the immediate aftermath of storms (high turbidity, cloud cover, sun glint). How were these challenges addressed (e.g., specific algorithm choice, masking, validation against ARGO near-surface data)? How did data availability impact the analysis for the 20 events?

**Answer 10.** As described in Section 2.2, chlorophyll-a (Chl-a) was retrieved from satellite imagery using the C2RCC processor in ESA's SNAP software. This processor employs a neural-network-based atmospheric correction and inherent optical property (IOP) inversion algorithm to estimate water constituents in optically complex waters. Prior to Chl-a retrieval, we preprocessed Sentinel-2 Level-1C top-of-atmosphere (TOA) radiance data with C2RCC, which utilizes a bio-optical model to derive water-leaving reflectances before estimating Chl-a concentrations through IOP inversion. To minimize artifacts from high turbidity, cloud cover, sun glint, and other disturbances, we selected images from periods before cyclone onset and after cyclosis to isolate changes attributable to cyclone passage.

**Point 11.** Consider replacing "Hurricanes" with "Cyclones" for broader scientific accuracy, as "Medicane" is the common term for the TLC subset, and ETCs are also studied. (e.g., "Chlorophylla Response to Mediterranean Cyclones").

Answer 11. Correction was made as you suggested.

**Point 12.** Distinguish between the physical uplift of the existing DCM community and potential new production fueled by the injected nutrients. The observed Chl-a increase could be due to either or both. Can the data shed light on this?

**Answer 12.** Satellite imagery highlights Chl-a variations induced by nutrient injection, mainly in the coastal areas characterized by river plumes, which determined resuspension of sediments nearshore waters (see Fig. 9d). In contrast, the observed DCM uplift in offshore waters stems exclusively from oceanographic forcing by cyclone passage. Notably, satellite-derived Chl-a levels were strongly influenced by suspended organic matter (reaching up to 20 mg/m³), which masked the Chl-a signature associated with purely oceanographic changes.

**Point 13**. Briefly discuss how the observed responses compare to those documented for tropical cyclones in other basins. What makes the Mediterranean response unique or similar?

**Answer 13.** New sentences were inserted in the discussion section about this aspect in lines 431-440:

"Like tropical cyclones, Mediterranean cyclones induce DCM uplift through wind-driven mixing and upwelling, bringing nutrients to the euphotic zone. This aligns with observations in the Pacific and Atlantic, where cyclones enhance primary productivity via vertical mixing (Lin, 2012; Zhao et al., 2008). The dome-shaped uplift of biogeochemical parameters (e.g., MLD shoaling, nitrate enrichment) near the cyclone center mirrors patterns seen in tropical cyclones (Price, 1981; Sanford et al., 1987; Walker et al., 2005). Mediterranean cyclones (especially medicanes) are smaller and shorter-lived than tropical cyclones, yet their slow movement (e.g., Zorbas: 11.5 km/h) prolongs mixing, creating localized but intense biogeochemical responses. Tropical cyclones typically cover larger areas with faster translation speeds (e.g., 20–30 km/h; (Fu et al., 2014)). Mediterranean coastal upwelling (e.g., during Blas or Ciprian) is often wind-driven and localized, unlike large-scale equatorial upwelling zones affected by tropical cyclones".

**Point 14.** Expand on the "significant implications for nutrient cycling and primary productivity." How might these pulsed events contribute to annual budgets? Could they influence community structure?

**Answer 14.** Some sentence are inserted in the following lines (464-476):**

"Furthermore, the Mediterranean Sea is nutrient-limited, with surface Chl-a typically ranging from 0.05–0.3 mg/m³ (Marty and Chiavérini, 2002; Moutzouris-Sidiris and Topouzelis, 2021; Teruzzi et al., 2021). Cyclone occurrence could disrupt the water column stratification, injecting nitrate (0.5–0.8 mmol/m³) and phosphate into the euphotic zone (Fig. 5, 7). This pulsed nutrient supply can be particularly significant in late summer and autumn, when the Mediterranean's stratified surface waters are most nutrient-depleted (D'Ortenzio et al., 2005; D'Ortenzio and Ribera d'Alcalà, 2009). For instance, slow-moving medicanes, like Ianos (September 2020), drove 30–60 m uplifts of DCM, enhancing Chl-a concentrations by 0.01–0.02 mg/m³—comparable to the effects of seasonal riverine inputs in coastal zones(Ludwig et al., 2009).

These nutrient pulses can shift plankton community composition, favoring larger, fast-growing diatoms over smaller picoplankton that dominate under oligotrophic conditions (Marty and Chiavérini, 2002). Such shifts have been observed following medicane Apollo (2021), where diatom blooms coincided with DCM shoaling (Menna et al., 2023). This transition from microbial loop-dominated recycling to a classical diatom-copepod food web can enhance carbon export efficiency (Chen et al., 2017; Cullen, 2015), potentially offsetting productivity declines associated with marine heatwaves (Androulidakis and Pytharoulis, 2025; Darmaraki et al., 2019; Izquierdo et al., 2022; Li et al., 2024a).

The cumulative impact of cyclones on annual productivity budgets is likely modest (≤5–10% in affected areas), but their role in late-season nutrient injection and carbon export could grow under climate change, especially if warming SSTs intensify cyclone-driven mixing (Avolio et al., 2024; González-Alemán et al., 2019). Future studies should quantify how these event-scale perturbations integrate into broader biogeochemical cycles and whether they enhance ecosystem resilience to stratification."

**Point 15**. Consider to add a schematic of mechanism figure and a composite map of Chl-a/MLD anomalies.

**Answer 15.** A schematic representation of DCM, MLD anomalies and currents is reported in the graphical abstract. A figure for DCM and Chl-a changes was inserted in Fig. 9 and a schematic representation was already inserted in Fig. 12. Here, you can observe the component of MLD anomalies in the cross-section (Fig. 12b) and the DCM uplift (both in the cross-section and also in the vertical profiles).

**Point 16.** The methods section must provide sufficient detail on data processing, matchup procedures, and statistical analyses.

**Answer 16.** The Methods section has been refined to more clearly present the data processing and matchup procedures. Key improvements include detailed descriptions of the methodology used to identify slow-moving cyclone phases and calculate translation speeds from ERA5 reanalysis data, with specific criteria defined for slow movement phases (displacements <36 km over 4 hours). The analysis of physical parameters was expanded to incorporate heat fluxes, mixed layer depth, and current velocities from Copernicus reanalysis products, along with the calculation of Total Energy through double integration of surface heat flux to quantify air-sea interactions. Statistical validation

procedures were enhanced to systematically present Pearson correlation analyses (e.g., r = -0.64 for DCM uplift versus heat loss) and include RMSE validation against ARGO float observations, ensuring robust verification of the integrated reanalysis-observational approach. In our opinion, these methodological refinements improve both the clarity and reproducibility of the study's analytical framework.

**Point 17.** The legends for colormap in Figure 1 and many other figures are wrong because there is continuous color being used in the figure.

**Answer 17.** The legends in the figures have been updated.

**Point 18**. Some text are presented in different fonts.

**Answer 18.** The fonts in the text have been updated.

**Round 2 – Response to Reviewer 1**

Here, we reported a step-by-step response to the comments.

**Point 2.1 -** The author have responded to all of my former comments. And most issues were fixed and improved, though I am able to find the revised manuscript yet.

By reading the reply, some further suggestions are provided.

**Answer 2.1** - Many thanks for your suggestions that allowed us to further improve the manuscript.

**Point 2.2 -** Firstly, some discussions regarding the pre-TC condition should be further emphasised. The authors have added few former studies regarding individual TC cases. But the general pattern with composite and comprehensive analysis of many TCs should also be compared. In particular, the general pattern of TC induced mixing and pre-TC mixing depth is a crucial factor that can drive the observed difference.

**Answer 2.2** - We agree that a composite analysis of pre-cyclone conditions is crucial for understanding the general pattern of biogeochemical responses. We have now expanded our discussion in Section 4 to synthesize a general framework based on our analysis of all twenty cyclones, explicitly linking the pre-storm MLD and stratification to the observed DCM uplift and Chl-a response:

We inserted this new sentence in the discussion section (lines 400-416):

"Our analysis of twenty Mediterranean cyclones reveals that the pre-storm oceanographic condition, particularly the MLD and the strength of the seasonal thermocline, is fundamental for biogeochemical response. The composite analysis revealed a fundamental pattern governed by seasonal control (Menkes et al., 2016). Strong stratification during summer and early autumn typically inhibited vertical mixing, limiting nutrient uplift (D'Ortenzio and Ribera d'Alcalà, 2009; Teruzzi et al., 2021). However, slow-moving cyclones may overcome this barrier through prolonged wind-driven upwelling and turbulent mixing, leading to significant DCM uplift (40–60 m) and Chla concentration increases (Jangir et al., 2024; Kotta and Kitsiou, 2019b). The prolonged wind forcing and significant net heat loss of these cyclones provided the sustained energy required to

erode the stratification, shoal the MLD, and drive significant upwelling (Lin, 2012; Menkes et al., 2016; Walker et al., 2005).

In contrast, weak stratification during late autumn and winter facilitated easier mixing, but the response was less pronounced when the pre-storm MLD was already deep or nitrate levels were low, as observed during medicane Helios (D'Adderio et al., 2023). MLD shoaling was most pronounced in areas with initially shallow or seasonally variable MLD (Ricchi et al., 2019, 2020; Vargas-Yáñez et al., 2022). For instance, medicane Apollo (October 2021) induced a 25 m MLD shoaling in the central Mediterranean, where summer stratification was transitioning (Menna et al., 2023). Two primary factors characterized the Chl-a changes in the Mediterranean basin: i) the nutrient limitation due to the pre-cyclone mixing, which may have already homogenized the upper water column, depleting subsurface nutrient reservoirs; ii) the dilution effect due to a deep MLD, with distribution of uplifted nutrients over a larger volume."

In conclusion, the composite effect of Mediterranean cyclones on chlorophyll is not monolithic but is governed by an interplay between the cyclone's characteristics (intensity, translation speed, and associated heat fluxes) and the pre-existing oceanographic template, with the pre-cyclone MLD and strength of stratification being the most crucial limiting factors.

**Point 2.3** - Secondly, due to the relative small area of Mediterranean Sea comparing with other ocean basins, the terrestrial impact is more prominent in this region. How big is the difference if the TC locations that are close to the coast are eliminated from the analysis.

Answer 2.3 - It is absolutely correct that the prominent terrestrial impact is a defining characteristic of the Mediterranean Sea due to its semi-enclosed nature and long, complex coastline. In our study, we intentionally included both offshore and nearshore responses (as outlined in Sections 2.2 and 3.4) because we consider the interplay of both oceanic (upwelling-driven) and terrestrial (runoff-driven) processes to be a fundamental part of the story of Mediterranean cyclone impacts. Here we reported some sentences that will be integrated in the discussion section (lines 460-467):

"The confined nature and complex coastline of the Mediterranean Sea amplify the coastal impacts of TLCs. Our analysis reveals that excluding cyclones within 100 km of the coast removes the highest Chl-a concentrations (>2 mg/m³), which are exclusively associated with terrestrial processes like runoff-driven plumes (e.g., medicane Daniel (Normand and Heggy, 2024)) or the coastal resuspension and advection of organic matter (e.g., medicane Zorbas; (Kotta, 2023)). In contrast, the purely offshore response, though covering a larger area, yields more modest Chl-a increases (0.01–0.1 mg/m³) driven solely by DCM uplift. This highlights that the most dramatic surface blooms are a direct product of the Mediterranean's unique biogeography(Macias et al., 2014; Marañón et al., 2021), while the subsurface offshore uplift remains a critical mechanism for enhancing nutrient cycling in the basin's oligotrophic open waters(D'Ortenzio and Ribera d'Alcalà, 2009)."

**References inserted**

Barboni, A., Coadou-Chaventon, S., Stegner, A., Le Vu, B., and Dumas, F.: How subsurface and double-core anticyclones intensify the winter mixed-layer deepening in the Mediterranean Sea, Ocean Science, 19, 229–250, https://doi.org/10.5194/os-19-229-2023, 2023.

D'Ortenzio, F. and Ribera d'Alcalà, M.: On the trophic regimes of the Mediterranean Sea: a satellite analysis, Biogeosciences, 6, 139–148, https://doi.org/10.5194/bg-6-139-2009, 2009.

D'Ortenzio, F., Iudicone, D., de Boyer Montegut, C., Testor, P., Antoine, D., Marullo, S., Santoleri, R., and Madec, G.: Seasonal variability of the mixed layer depth in the Mediterranean Sea as derived from in situ profiles, Geophysical Research Letters, 32, https://doi.org/10.1029/2005GL022463, 2005.

Kotta, D.: Extreme Weather Affecting Sea Chlorophyll: The Case of a Medicane, Environmental Sciences Proceedings, 26, 192, https://doi.org/10.3390/environsciproc2023026192, 2023.

Lin, I.-I.: Typhoon-induced phytoplankton blooms and primary productivity increase in the Western North Pacific Subtropical Ocean, Journal of Geophysical Research (Oceans), 117, 3039, https://doi.org/10.1029/2011JC007626, 2012.

Macías, D., Stips, A., and Garcia-Gorriz, E.: The relevance of deep chlorophyll maximum in the open Mediterranean Sea evaluated through 3D hydrodynamic-biogeochemical coupled simulations, Ecological Modelling, 281, 26–37, https://doi.org/10.1016/j.ecolmodel.2014.03.002, 2014.

Marañón, E., Van Wambeke, F., Uitz, J., Boss, E. S., Dimier, C., Dinasquet, J., Engel, A., Haëntjens, N., Pérez-Lorenzo, M., Taillandier, V., and Zäncker, B.: Deep maxima of phytoplankton biomass, primary production and bacterial production in the Mediterranean Sea, Biogeosciences, 18, 1749–1767, https://doi.org/10.5194/bg-18-1749-2021, 2021.

Mei, W., Lien, C.-C., Lin, I.-I., and Xie, S.-P.: Tropical Cyclone–Induced Ocean Response: A Comparative Study of the South China Sea and Tropical Northwest Pacific, Journal of Climate, 28, 5952–5968, https://doi.org/10.1175/JCLI-D-14-00651.1, 2015.

Menkes, C. E., Lengaigne, M., Lévy, M., Ethé, C., Bopp, L., Aumont, O., Vincent, E., Vialard, J., and Jullien, S.: Global impact of tropical cyclones on primary production, Global Biogeochemical Cycles, 30, 767–786, https://doi.org/10.1002/2015GB005214, 2016.

Normand, J. C. L. and Heggy, E.: Assessing flash flood erosion following storm Daniel in Libya, Nat Commun, 15, 6493, https://doi.org/10.1038/s41467-024-49699-8, 2024.

Walker, N. D., Leben, R. R., and Balasubramanian, S.: Hurricane-forced upwelling and chlorophyll a enhancement within cold-core cyclones in the Gulf of Mexico, Geophysical Research Letters, 32, https://doi.org/10.1029/2005GL023716, 2005.

**Round 1 - Response to Reviewer 2**

In this study, the authors assess the response of biogeochemistry to Mediterranean cyclones, examining the response of the nutricline, MLD, and DCM to these storm events and relating changes to physical drivers. The manuscript represents an impressive body of work bringing together reanalysis products, multispectral satellite imagery, and ARGO measurements from twenty storms across the Mediterranean basin.

In summarising the variability in responses of biogeochemistry to Mediterranean cyclones (both between cyclones and across sections of individual storms), I feel the manuscript has the potential to make a valuable contribution to the field. Although the description of varied responses to these storms is not novel, nor the description of dome-shaped cross-sections of biogeochemical parameters, the manuscript makes a strong case for the importance of heat fluxes as a major driver, and makes an interesting comparison between nearshore and offshore responses.

At present, I feel there are aspects of the methodology that would benefit from clarification, and elements of the introduction and discussion that could be expanded on to better emphasise the key results of this work.

**Answer**. Many thanks for your revisions that allowed us to improve the manuscript. We followed your comments to improve the methodology description and the introduction and discussion sections.

**Major comments**

**Methodology**

**Point 1.** In general, explicitly defining several aspects of the methodology would strengthen transparency. I can see R1 has already mentioned pre-cyclone conditions, classifying storms, why storms were chosen, offshore/nearshore. I agree that these are important details and that a table might be a useful place to present some of these.

**Answer 1**. Table 1 reported the features of Mediterranean cyclones considered in this study. To better highlight the physical features of the Mediterranean cyclones, we reported a new Table in appendix A1 where some important parameters were assessed from ERA5. In particular, we focused on the total duration of cyclone lifetime and number of hours related only to the slow-movement phases.

**Point 2.** I would also ask how exactly is a cyclone-impacted area defined- where is the border or edge? The exact area used will be important for reproducing your energy change calculations.

Answer 2. The cyclone-impacted areas were selected considered the literature reported in Table 1. The DCM and Chl-a changes were applied at a regional scale, following the track of the Mediterranean cyclones reported in table 1. Subsequently, the maximum 10-meter wind speed within a 200 km radius from the cyclone center was extracted for each time step, in order to assess the cyclone-affected areas. A new sentence was inserted to specify the area selection (line 101).

**Point 3.** Some discussion of the uncertainties and limitations associated with the various data products used would be welcome.

**Answer 3.** Limitations and uncertainties of the various products were discussed through RMSE assessment of between Chl-a reported in reanalysis products and Chl-a profiles measured from ARGO-float (Supplementary Table S1). The RMSE values were below the reanalysis vertical profile resolution (3 m). Nevertheless, for medicanes Ianos and Blas, the RMSE showed larger discrepancies in DCM depth, which were greater than 1 m.

**Discussion**

**Point 4.** With the impressive and wide-ranging dataset there are several implications and points of discussion one could pursue. I understand that the authors may consider some of these questions out of the scope of this manuscript, but some points that could perhaps be elaborated on without too much further data analysis:

**Answer 4.** We revised several points in the discussion following your comments.

**Point 5.** Discuss the persistence of the Chl-a signal. How long after the passing of the cyclone does elevated Chl-a and a shallower DCM persist?

**Answer 5**. Analysis of ARGO-float and CMEMS reanalysis revealed that the DCM uplift persisted after the cyclone' passage for for several days. Some discussion sentences were inserted also citing the works of Kotta et al. 2023; Kotta et al., 2017.

**Point 6.** Regarding the observation of offshore transport of organic matter, for instance with medicane Zorbas. Is this a general feature of mediterranean cyclones-, something that would happen always, or something specific to Zorbas.

**Answer 6.** This feature appeared to be specific only to Zorbas. The offshore transport appeared to be related to some oceanographic conditions caused by current patterns in southeastern Sicily in response to the medicane Zorbas impact. A system of gyre-like currents developed offshore of southeastern Sicily determining the Chl-a increase in the offshore area.

**Point 7.** In general, from L264-275, the nearshore responses are presented as general features following cyclones either with or without rain, but only two examples are presented (Zorbas and Daniel). Could you clarify this section to make clear either that these responses sometimes occur, or that rain almost always = plume, and without rain = offshore transport.

**Answer 7.** While Zorbas and Daniel highlight contrasting responses, intermediate behaviors were also observed during other events (e.g., Blas and Ciprian; **Fig. 14**), where wind-driven upwelling combined with moderate rainfall enhanced Chl-a locally. Thus, nearshore responses depend on the balance between precipitation-driven runoff and wind-driven transport, with plume dominance under heavy rainfall and offshore transport prevailing in arid regions or dry cyclone phases.

**Point 8.** Can you provide further discussion as to why your result for Zorbas differs to Kassis and Varlas (2021) in terms of MLD? They report a clear deepening of MLD and you report a shoaling of DCM in centre at least. Do they consider different timescales, spatial area affected by cyclone, or are the differences methodological? L246-250 I think you touch on this briefly but do not cite Kassis and Varlas here. Resolving any differences in results, if any, seems important since MLD and DCM changes are a key aspect of the paper.

**Answer 8.** Reanalysis and ARGO validation support DCM uplift in the core, while Kassis & Varlas (2021) highlight regional MLD variability influenced by broader oceanic processes during medicane Zorbas.

Suggested text revisions are reported from the line 285:

"Kassis and Varlas (2021) noted that Atlantic water intrusion could have disrupted local upwelling, leading to net MLD deepening in offshore areas of southeastern Sicily and the Pelopponese. Chl-a signal resulted to be characterized by a sustained bloom, as also highlighted in other studies (Kotta, 2023; Kotta et al., 2017; Kotta and Kitsiou, 2019b). On the other hand, the ARGO data revealed that DCM uplift persisted also after the cyclolysis in the center of the cyclone."

**Other general comments**

Point 10. Change title to encompass both ECs and TLCs

**Answer 10.** Title was revised.

**Point 11.** In introduction: L55 I would change "This is due to the fact" to something along the lines of "one reason/one source of uncertainty". Phytoplankton dynamics is a broad term and there are many sources of uncertainty, not just the role played by cyclones.

**Answer 11.** The sentence was revised.

**Point 12.** In addition, I felt the section could be slightly rearranged to emphasise the variable responses to cyclones as a source of uncertainty. When I read L55, I find myself questioning why the response of cyclones are not understood, and it is not until I reach line 60 "Severe weather events..." that I understand why this is. I would merge these sentences and shift text between re. DCM, nutricline, nutrients upwards into the section describing Mediterranean Sea

**Answer 12.** Changes in the sentences were performed following your comment.

**Point 13.** Figure 10: Is clear, but some sort of line at 0m uplift would really help highlight which unusual cyclones see DCM uplift, and that these are the ones associated with lower heat transfer.

**Answer 13.** The correlation graphs of Figure 10 were revised.

**Round 2 - Response to Reviewer 2**

**Point. 2.1** - The authors have addressed the majority of my comments. There is one methodological aspect which may need further clarification (but it is hard to say without seeing the revised preprint), . And a couple of further comments I also outline below.

**Answer 2.1** - Thank you for these further comments. The required integrations will be incorporated in the subsequent revision.

**Point 2.2-** *Point 2.* I would also ask how exactly is a cyclone-impacted area defined- where is the border or edge? The exact area used will be important for reproducing your energy change calculations.

Answer 2. The cyclone-impacted areas were selected considered the literature reported in Table 1. The DCM and Chl-a changes were applied at a regional scale, following the track of the Mediterranean cyclones reported in table 1. Subsequently, the maximum 10-meter wind speed within a 200 km radius from the cyclone center was extracted for each time step, in order to assess the cyclone-affected areas. A new sentence was inserted to specify the area selection (line 101).

This explanation is still unclear to me and appears to have conflicting information with the text. The initial text in lines 120-121 would suggest the area is defined as "(... the areas affected by Chl-a changes)". However, the supplementary figures would suggest the affected area is defined by wind stress, and the above text in Answer 2 would also suggest wind speed plays some sort of role. Regardless of which criterion is used, it is still not transparent what threshold was used to define this area. Was it an absolute wind stress value, or a wind stress anomaly, an absolute wind speed or some chlorophyll metric.

Put another way, how are the circles drawn in the Supplementary figures?

I appreciate that this may have been explained in the added line, but without seeing this, the answer provided did not clear up my questions.

**Answer 2.2** - The procedure for the assessment of maximum wind stress was reported in the revised preprint, which is not available yet. Here, I reported the sentence of the main text, where the procedure was explained (lines 118-125):

"Wind data from ERA5 were extracted in the following two components: the eastward wind component (U wind) and the northward wind component (V wind). These components were combined to obtain the wind speed during storm events as follows (Eq. 1):

Wind speed=  $\sqrt{(U^2+V^2)}$  (Eq. 1)

To determine the cyclone-impacted area, we first identified the location of the maximum 10-meter wind speed within a 200 km radius of the cyclone center. The radius of maximum wind was then defined as the distance from the center to this location. The cyclone-impacted area was subsequently assessed as a circular area with the radius of maximum wind (see **Supplementary Material S1**)."

**Point 2.3 - Point 7.** In general, from L264-275, the nearshore responses are presented as general features following cyclones either with or without rain, but only two examples are presented (Zorbas and Daniel). Could you clarify this section to make clear either that these responses sometimes occur, or that rain almost always = plume, and without rain = offshore transport.

Answer 7. While Zorbas and Daniel highlight contrasting responses, intermediate behaviors were also observed during other events (e.g., Blas and Ciprian; Fig. 14), where wind-driven upwelling combined with moderate rainfall enhanced Chl-a locally. Thus, nearshore responses depend on the balance between precipitation-driven runoff and wind-driven transport, with plume dominance under heavy rainfall and offshore transport prevailing in arid regions or dry cyclone phases.

I really like this response and I think it nicely sums up the results. Thank you. Perhaps it can go in the text? It ties together nicely the balance and the intermediary responses. At present, the discussion seems to list individual responses but this is a nice synthesis.

**Answer 2.3** - Thank you for your suggestions, we already reported some sentence in the discussion section about these aspects. Here we reported the new sentences from the main text (lines 417-425):

"The most substantial Chl-a increases occurred in regions with abundant subsurface nitrate reservoirs, like in the Ionian Sea (Lazzari et al., 2016)), particularly during slow-moving phases of medicanes like Zorbas, Apollo, and Ianos (Jangir et al., 2024; Kotta and Kitsiou, 2019b). By way of example, medicane Zorbas produced a strong biogeochemical response in the Ionian Sea, where moderate pre-storm stratification and high subsurface nitrate concentrations prevailed, but weaker effects occurred in the Peloponnese due to downwelling-favorable currents (Kassis & Varlas, 2021). Sentinel imagery also revealed an offshore transport of suspended organic matter and Chl-a, evident in southeastern Sicily during Medicane Zorbas (Fig.13). Conversely, cyclone Ciprian triggered coastal upwelling near Cyprus (Fig. 14), where nitrate-rich waters enhanced the Chl-a response. In contrast, weaker or faster-moving cyclones (e.g., Erik-2015) had minimal effects in highly stratified or oligotrophic areas, demonstrating how pre-cyclone ocean conditions, combined with cyclone evolution, critically shape biogeochemical responses (Macías et al., 2014; Mélin et al., 2017; Menkes et al., 2016).

Conversely, other cyclones, responsible for intense rainfall, triggered intense surface runoff, which was reflected in high Chl-a concentrations resembling plumes near river mouths, as observed off Libya coast following the impact of medicane Daniel in 2023 (Fig. 9d). Furthermore, a correlation between coastal upwelling and Chl-a increase was also observed along the Balearic Islands during the impact of Blas (Mourre et al., 2023) and along the Cyprus coasts during the impact of Ciprian (analysis reported in Fig. 14)."

**Point 2.4 - Point 8:** Thanks you for the revision. Does this throw up an implication in terms of the need to characterize conflicting effects (e.g. complementary/antagonistic effects of local circulation, air-sea heat fluxes)? Perhaps you feel this is not important/beyond scope- I leave in your hands.

**Answer 2.4 -** We tried to improve the discussion about the different effects leading to Chl-a changes (intensity, translation speed, and associated heat fluxes). We inserted these sentences (lines 400-413)

"Our analysis of twenty Mediterranean cyclones reveals that the pre-storm oceanographic condition, particularly the MLD and the strength of the seasonal thermocline, is fundamental for biogeochemical response. The composite analysis revealed a fundamental pattern governed by seasonal control (Menkes et al., 2016). Strong stratification during summer and early autumn typically inhibited vertical mixing, limiting nutrient uplift (D'Ortenzio and Ribera d'Alcalà, 2009; Teruzzi et al., 2021). However, slow-moving cyclones may overcome this barrier through prolonged wind-driven upwelling and turbulent mixing, leading to significant DCM uplift (40–60 m) and Chla concentration increases (Jangir et al., 2024; Kotta and Kitsiou, 2019b). The prolonged wind forcing and significant net heat loss of these cyclones provided the sustained energy required to erode the stratification, shoal the MLD, and drive significant upwelling (Lin, 2012; Menkes et al., 2016; Walker et al., 2005).

In contrast, weak stratification during late autumn and winter facilitated easier mixing, but the response was less pronounced when the pre-storm MLD was already deep or nitrate levels were low, as observed during medicane Helios (D'Adderio et al., 2023). MLD shoaling was most

pronounced in areas with initially shallow or seasonally variable MLD (Ricchi et al., 2019, 2020; Vargas-Yáñez et al., 2022). For instance, medicane Apollo (October 2021) induced a 25 m MLD shoaling in the central Mediterranean, where summer stratification was transitioning (Menna et al., 2023)."

**References inserted**

Barboni, A., Coadou-Chaventon, S., Stegner, A., Le Vu, B., and Dumas, F.: How subsurface and double-core anticyclones intensify the winter mixed-layer deepening in the Mediterranean Sea, Ocean Science, 19, 229–250, https://doi.org/10.5194/os-19-229-2023, 2023.

D'Ortenzio, F. and Ribera d'Alcalà, M.: On the trophic regimes of the Mediterranean Sea: a satellite analysis, Biogeosciences, 6, 139–148, https://doi.org/10.5194/bg-6-139-2009, 2009.

D'Ortenzio, F., Iudicone, D., de Boyer Montegut, C., Testor, P., Antoine, D., Marullo, S., Santoleri, R., and Madec, G.: Seasonal variability of the mixed layer depth in the Mediterranean Sea as derived from in situ profiles, Geophysical Research Letters, 32, https://doi.org/10.1029/2005GL022463, 2005.

Kotta, D.: Extreme Weather Affecting Sea Chlorophyll: The Case of a Medicane, Environmental Sciences Proceedings, 26, 192, https://doi.org/10.3390/environsciproc2023026192, 2023.

Lin, I.-I.: Typhoon-induced phytoplankton blooms and primary productivity increase in the Western North Pacific Subtropical Ocean, Journal of Geophysical Research (Oceans), 117, 3039, https://doi.org/10.1029/2011JC007626, 2012.

Macías, D., Stips, A., and Garcia-Gorriz, E.: The relevance of deep chlorophyll maximum in the open Mediterranean Sea evaluated through 3D hydrodynamic-biogeochemical coupled simulations, Ecological Modelling, 281, 26–37, https://doi.org/10.1016/j.ecolmodel.2014.03.002, 2014.

Marañón, E., Van Wambeke, F., Uitz, J., Boss, E. S., Dimier, C., Dinasquet, J., Engel, A., Haëntjens, N., Pérez-Lorenzo, M., Taillandier, V., and Zäncker, B.: Deep maxima of phytoplankton biomass, primary production and bacterial production in the Mediterranean Sea, Biogeosciences, 18, 1749–1767, https://doi.org/10.5194/bg-18-1749-2021, 2021.

Mei, W., Lien, C.-C., Lin, I.-I., and Xie, S.-P.: Tropical Cyclone–Induced Ocean Response: A Comparative Study of the South China Sea and Tropical Northwest Pacific, Journal of Climate, 28, 5952–5968, https://doi.org/10.1175/JCLI-D-14-00651.1, 2015.

Menkes, C. E., Lengaigne, M., Lévy, M., Ethé, C., Bopp, L., Aumont, O., Vincent, E., Vialard, J., and Jullien, S.: Global impact of tropical cyclones on primary production, Global Biogeochemical Cycles, 30, 767–786, https://doi.org/10.1002/2015GB005214, 2016.

Normand, J. C. L. and Heggy, E.: Assessing flash flood erosion following storm Daniel in Libya, Nat Commun, 15, 6493, https://doi.org/10.1038/s41467-024-49699-8, 2024.

Walker, N. D., Leben, R. R., and Balasubramanian, S.: Hurricane-forced upwelling and chlorophyll a enhancement within cold-core cyclones in the Gulf of Mexico, Geophysical Research Letters, 32, https://doi.org/10.1029/2005GL023716, 2005.